



# High-resolution fully-coupled atmospheric–hydrological modeling: a cross-compartment regional water and energy cycle evaluation

Benjamin Fersch[1], Alfonso Senatore[2], Bianca Adler[3], Joël Arnault[1], Matthias Mauder[1], Katrin Schneider[1], Ingo Völksch[1], and Harald Kunstmann[1]

[1]Karlsruhe Institute of Technology, Institute of Meteorology and Climate Research (IMK-IFU), Garmisch-Partenkirchen, Germany
[2]University of Calabria, Department of Environmental and Chemical Engineering, Rende (CS), Italy
[3]Karlsruhe Institute of Technology, Institute of Meteorology and Climate Research (IMK-TRO), Karlsruhe, Germany

*Correspondence to:* Benjamin Fersch (fersch@kit.edu)

**Abstract.** The land surface and the atmospheric boundary layer are closely intertwined with respect to the exchange of water, trace gases and energy. Nonlinear feedback and scale dependent mechanisms are obvious by observations and theories. Modeling instead is often narrowed to single compartments of the terrestrial system or largely bound to traditional disciplines. Coupled terrestrial hydrometeorological modeling systems attempt to overcome these limitations to achieve a better integra-

tion of the processes relevant for regional climate studies and local area weather prediction. This study examines the ability of the hydrologically enhanced version of the Weather Research and Forecasting Model (WRF-Hydro) to reproduce the regional water cycle by means of a two-way coupled approach and assesses the impact of hydrological coupling with respect to a traditional regional atmospheric model setting. It includes the observation-based calibration of the hydrological model component (offline WRF-Hydro) and a comparison of the classic WRF and the fully coupled WRF-Hydro models both with

identical calibrated parameter settings for the land surface model (Noah-MP). The simulations are evaluated based on extensive observations at the preAlpine Terrestrial Environmental Observatory (TERENO-preAlpine) for the Ammer ($600\,\mathrm{km}^2$) and Rott ($55\,\mathrm{km}^2$) river catchments in southern Germany, covering a five month period (Jun–Oct 2016). The sensitivity of 7 land surface parameters is tested using the *Latin-Hypercube One-factor-At-a-Time* (LH-OAT) method and 6 sensitive parameters are subsequently optimized for 6 different subcatchments, using the Model-Independent *Parameter Estimation and Uncertainty*

*Analysis software* (PEST). The calibration of the offline WRF-Hydro gives Nash-Sutcliffe efficiencies between 0.56 and 0.64 and volumetric efficiencies between 0.46 and 0.81 for the six subcatchments. The comparison of classic WRF and fully coupled WRF-Hydro, both using the calibrated parameters from the offline model, shows nominal alterations for radiation and precipitation but considerable changes for moisture- and heat fluxes. By comparison with TERENO-preAlpine observations, the fully coupled model slightly outperforms the classic WRF with respect to evapotranspiration, sensible and ground heat

flux, near surface mixing ratio, temperature, and boundary layer profiles of air temperature. The subcatchment-based water budgets show uniformly directed variations for evapotranspiration, infiltration excess and percolation whereas soil moisture and precipitation change randomly.



## 1 Introduction

The spatial distribution of land surface properties is a key component for regional weather and climate manifestation. Contrary to constant or slowly varying features like topography or landcover, hydrological processes contribute vastly to the spatio-temporal dynamics that influence the exchange of water and energy at the land surface–atmosphere interface.

Earth system and climate models could greatly profit from more sophisticated descriptions of hydrological processes (Clark et al., 2015). Moreover, as shown by Ning et al. (2019), in scientific literature, the topic of coupled hydrological–atmospheric modeling is constantly gaining popularity. Several physically based, fully coupled hydrological–atmospheric models have been developed by the scientific community over the past 15 years, addressing non-linear cross-compartment feedback and fostering a closed representation of regional water and energy cycles (e.g., Shrestha et al., 2014; Butts et al., 2014; Gochis

et al., 2016; Soltani et al., 2019). Comprehensive reviews on the history of fully coupled hydrometeorological models and their application can be found in Wagner et al. (2016), Senatore et al. (2015), and Ning et al. (2019). Typically, these models are amalgamations of preexisting subject-specific algorithms of varying complexity, with land surface models being the common thread. Recent applications of fully coupled models show promising results in improving spatial pattern dynamics and area integrals of regional water budgets. However, the research field is far away from maturity and many further studies are required.

Using ParFlow coupled with the Community Land Model, Maxwell and Kollet (2008) found that for the U.S. Oklahoma southern Great Planes, groundwater depth governs the sensitivity of regions to variations in temperature and precipitation. Larsen et al. (2016) reported that by accounting for shallow groundwater in the fully coupled model MIKE SHE, summer evapotranspiration results improved for a study over Kansas, USA. While several studies highlight the importance of lateral hydrological processes for the improved simulation of soil moisture (e.g., Wagner et al., 2016; Larsen et al., 2016), the sensitiv-

ity for land subsurface–surface–planetary boundary layer feedback and precipitation generation is less pronounced, especially for the humid regions with strong synoptic forcing (e.g., Butts et al., 2014; Barlage et al., 2015; Arnault et al., 2018; Rummler et al., 2018; Sulis et al., 2018).

    Coupled modeling studies often focus on single objective variables for validation (like, e.g., discharge, evapotranspiration or soil moisture) or restrict their analysis to describing only the changes in simulation results without any comparison to

observations. Targeting single variables can result in the problem of equifinality where model realizations are proven skillful for a single aspect, yet possibly being wrong for several others. To investigate if a certain model or model configuration can provide improved realism, the limited perspective of single or few variable evaluations needs to be abandoned (García-Díez et al., 2015). To overcome the dilemma, fully coupled simulations should be validated and evaluated with respect to as many independent observations as possible. However, the scales of simulations and observations need to match. For catchment-

scale coupled hydrometeorological models, most of the global data products (e.g., from satellite) are rather coarse. Regional observatories with integrative measurements of the subsurface to boundary layer fluxes and states provide a sound basis for a holistic evaluation. In the recent past, several efforts have been undertaken to create comprehensive observation sets that allow for subsurface to atmosphere integrated studies of water and energy fluxes for small to medium scale river catchments. The most prominent activities for Europe are HOAL (Hydrological Open Air Observatory, Blöschl et al., 2016), HOBE (The



Danish Hydrological Observatory, Jensen and Illangasekare, 2011), LAFO (Land-atmosphere feedback observatory, Spath et al., 2018), and TERENO (TERrestrial ENvironmental Observatories, Zacharias et al., 2011). Although two of them address hydrology in their names, land–atmosphere interaction is a central research item for all of these observatories.

In this study, we evaluate the effect of bidirectional hydrological–atmospheric model coupling with respect to 1) the land sur-
face energy flux partitioning and 2) to the different compartments of the hydrological cycle. To that end, we perform uncoupled and fully coupled simulations with the hydrologically enhanced version of the Weather Research and Forecasting modeling system (WRF-Hydro, Gochis et al., 2016) for the Ammer river catchment region, located in southern Bavaria, Germany. We utilize convection resolving resolution of $1 \, km^2$ for the atmospheric part together with a 100 by 100 m hydrological subgrid. For validation, we employ a rich and comprehensive dataset consisting of observations from the TERENO-preAlpine observa-
tory (Kiese et al., 2018), enhanced by data from the ScaleX field campaign (Wolf et al., 2016), complemented by further local providers.

## 2   Methods and data

### 2.1   Study area

The study area covers the two medium sized river catchments Ammer ($600 \, km^2$) and Rott ($55 \, km^2$), located in southern Bavaria,
Germany (Fig. 1). The hydromorphic characteristics of this Alpine front-range region were formed during the last glacial and predominantly feature Gley- Cambi-, and Histosols on top of carbon based gravel deposits. Elevations range from above 2300 m ASL in the south, down to 533 m ASL at the outlet towards lake Ammer. Landcover is dominated by meadows and forests. The proportion of forests rises from about 20 % in the north to 57 % in the Alpine part of the catchment (Fetzer et al., 1986). Due to the climatic conditions, crops are only of minor importance and are only prevalent in the lower part. Mean annual precipitation
exhibits a gradient from 950 mm close to lake Ammer to more than 2000 mm in the mountains. Mean annual evapotranspiration shows no distinct correlation with elevation and ranges from 300 mm at the sparsely vegetated mountain slopes to 500–600 mm for the rest of the catchment. Mean annual temperature ranges from about 7 to 4 °C between the lower and the upper parts of the catchment.

The study region is part of the German Terrestrial Environmental Observatory program (TERENO, Zacharias et al., 2011),
an initiative for the long-term monitoring of climate environmental variables. Our study is bound to the multidisciplinary field experiment and observation campaign *ScaleX* (Wolf et al., 2016) that took place at the TERENO-preAlpine DE-Fen site (Fig. 1), in the summers of 2015 and 2016.

### 2.2   Observation data

The study region was selected to cover the Helmholtz preAlpine TERrestrial ENvironmental Observatory (TERENO-preAlpine)
located in the foothills of the Bavarian Alps of southern Germany. TERENO-preAlpine features observations for the range of compartments of the terrestrial hydrometeorological cycle. The observatory has been designed for long-term monitoring of

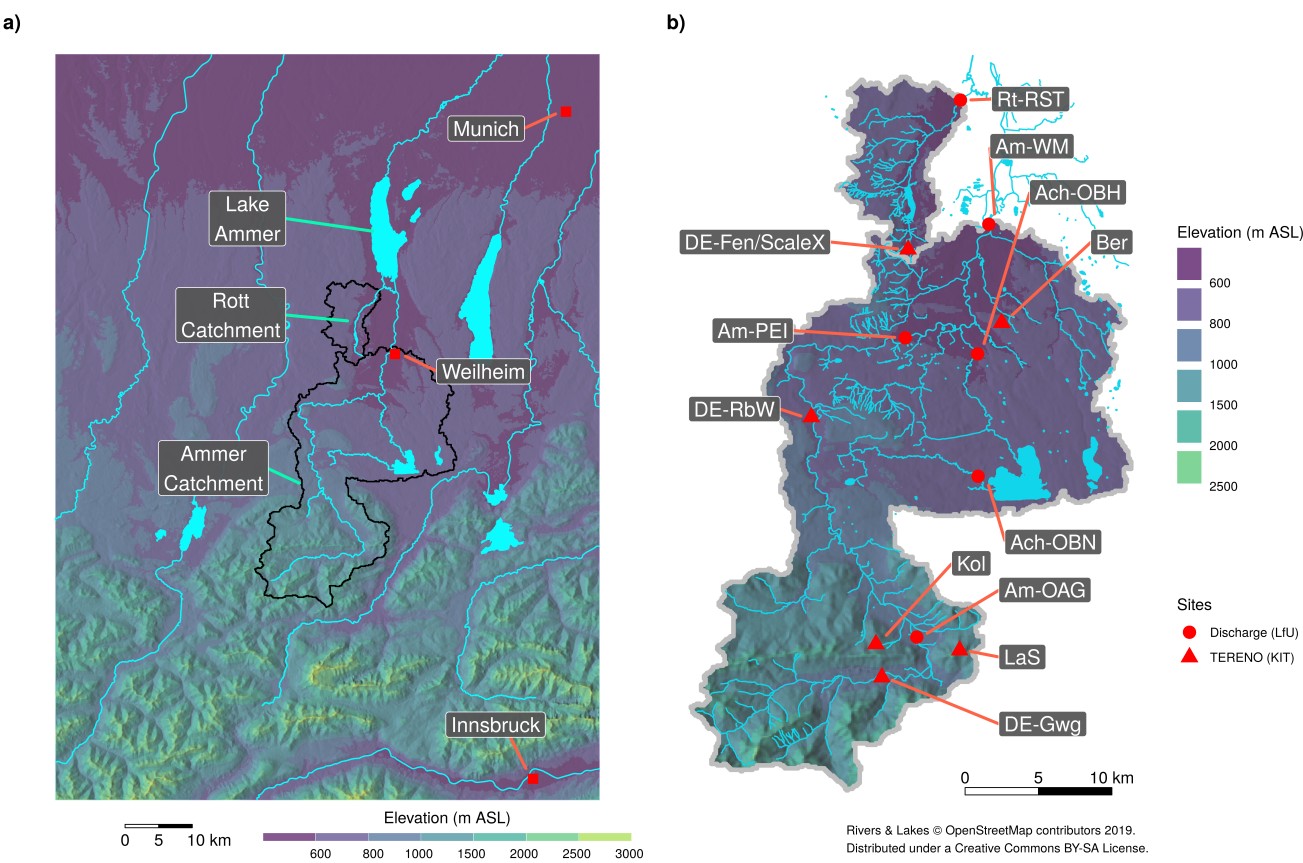

**Figure 1.** a) elevation map for model domain 3 and location of Ammer and Rott river catchments. b) catchment detail and observation locations. Abbreviations – river gauges maintained by the Bavarian Environmental Agency (LfU): Rt-RST = Rott-Raisting, Am-WM = Ammer-Weilheim, Am-PEI = Ammer-Peißenberg, Am-OAG = Ammer-Oberammergau, Ach-OBH = Ach-Oberhausen, Ach-OBN = Ach-Obernach; TERENO preAlpine sites: DE-Fen = Peißenberg-Fendt, Ber = Oberhausen-Berg, DE-RbW = Rottenbuch, Kol = Oberammergau-Kolbensattel (mountain station), Las = Oberammergau-Laber (mountain station), DE-Gwg = Ettal-Graswang. Elevation is derived from ASTER GDEM.





climatological and ecological variables. A detailed description of the concept is available in Kiese et al. (2018). Figure 1b provides an overview of the measurement sites that comprise standard climatological, eddy-covariance, lysimeter, soil moisture, groundwater and discharge observations.

The observed sensible and latent heat fluxes presented in this study are determined by means of tower-based eddy-covariance measurements, which are operated on a long-term basis. These installations comprise a CSAT3 sonic anemometer (Campbell Scientific Inc., Logan, UT) at the three main sites (DE-Fen, DE-RbW, DE-Gwg) of TERENO-preAlpine and a LI-7500 infrared gas analyser at DE-Fen and DE-Gwg, while DE-RbW is equipped with a LI-7200 gas analyser (Licor Biosciences Inc., Lincoln, NE). The measurement height of these systems is 3.3 m above ground. High-frequency data from these instruments are recorded digitally on a Campbell CR3000 data logger. The fluxes are computed in the field every day with an industrial PC using the eddy-covariance software TK3 (Mauder and Foken, 2015), including corrections for misalignment of the anemometer using the double rotation methods (Wilczak et al., 2001), humidity influences on the sonic temperature measurement (Schotanus et al., 1983), spectral losses due to path averaging and sensor separation (Moore, 1986), and density fluctuations (Webb et al., 1980). Automated quality control and uncertainty assessment is applied in accordance with Mauder et al. (2013), which extends the test of Foken and Wichura (1996) by an additional spike test on the high-frequency data, a test on the interdependence of fluxes due to the flux corrections and a test on the representativeness of the flux footprint (Kormann and Meixner, 2001). Moreover, an energy-balance-closure-adjustment method, which is based on the daily energy balance ratio is applied to daytime sensible and latent heat flux data under the condition that the Bowen ratio is preserved (Mauder et al., 2013).

Lysimeter data is available for three of the TERENO-preAlpine sites (DE-Fen, DE-RbW, DE-Gwg). The measurements are separated for representative treatments of extensive and intensive grassland management, in accordance with the local farmer's cutting and fertilizer management (Fu et al., 2017). For this study, data derived from six control lysimeters per site are taken into account (i.e., lysimeters that were excavated at adjacent grassland sites nearby the experimental site). For each lysimeter, precipitation, evapotranspiration and groundwater recharge (percolation) is calculated from the variations in total weight and the changes in water volume of the corresponding water tank. Obvious outliers in the weight measurements are removed above thresholds of $1000\,\text{g min}^{-1}$ and $200\,\text{g min}^{-1}$ for the weight changes of the lysimeters and water tanks, respectively. Furthermore, for separation of signal and noise, the *Adaptive Window and Adaptive Threshold filter* (AWAT, Peters et al., 2014) is applied to the time series of weight changes of each individual lysimeter and corresponding water tank at a temporal resolution of 1 min. The procedure applied in this study is further described by Fu et al. (2017).

A wireless sensor network at the DE-Fen site, consisting of 55 profiles (5, 20, 50 cm), provides soil moisture information for a grassland area of roughly 12 ha. The measurement devices are spade-shaped ring oscillator electromagnetic permittivity sensors (Truebner SMT 100, Bogena et al., 2017) with a vertical representativeness of about 3 cm. Additional information on sensor calibration and the conversion of permittivity into volumetric water content is available in Fersch et al. (2018).

Within the course of the ScaleX campaign (June-August 2016, Wolf et al., 2016), a scanning microwave radiometer (HAT-PRO, Humidity and temperature profiler, Rose et al., 2005) provided information on temperature and humidity profiles as well as integrated water vapor and liquid water path. The instrument measures sky brightness temperature at 14 frequencies, seven are distributed between 22.235 and 31.4 GHz along the wing of the 22.235 water vapor line and seven between 51.26





and 58 GHz along the wing of the 60 GHz oxygen absorption complex. Information on atmospheric variables is obtained from the measured brightness temperatures with a retrieval algorithm from the University of Cologne (Löhnert and Crewell, 2003; Löhnert et al., 2009). For the retrieval creation a set of around 14000 radiosonde profiles, measured at Munich (station at 489 m ASL) between 1990 and 2014, is used. While integrated water vapor has an accuracy of less than 1 kg m$^{-2}$ (Pospichal

and Crewell, 2007), the vertical resolution of humidity is low as only two of the seven available water vapor channels are independent (Löhnert et al., 2009). Adding the information from 9 elevation scans performed at low angles to the standard zenith observations for the opaque oxygen complex allows to obtain temperature profiles with a higher spatial resolution and an accuracy of less than 1 K below around 1.5 km (Crewell and Löhnert, 2007).

Discharge measurements for the six subcatchments (Ach-OBN, Ach-OBH, Am-OAG, Am-PEI, Am-WM, Rt-RST) evalu-
ated in this study are obtained from the online archive of the Bavarian Environmental Agency (LfU, https://www.gkd.bayern.de).

## 2.3  The WRF-Hydro modeling system

The Weather Research and Forecast modeling system (WRF, Skamarock and Klemp, 2008) is a common, community developed tool for the simulation of local area to global tropospheric dynamics and their interaction with the land surface. Applications range from short-term regional forecast to long-term continental climate studies with spatial resolutions of a few tens of meters
with large-eddy simulations to several kilometers. WRF-Hydro (Gochis et al., 2016) augments WRF with respect to lateral hydrological processes at and below the land surface. It features one-way and two-way coupling of local area atmospheric and hydrological models and comes with several process-based hydrological modules already implemented (i.e., for river channel, overland and subsurface flow routing, baseflow estimation). The one-way coupled WRF-Hydro system (i.e., separate computations for atmosphere and hydrology without upward feedback) had been successfully applied for short-term forecasting
(Yucel et al., 2015) and long-term hindcasting (Li et al., 2017) and was furthermore selected as the core component of the United States National Water Model (NWM, e.g., Cohen et al., 2018, http://water.noaa.gov/about/nwm). Fully, two-way coupled applications found reasonable performance for monthly scale discharge simulations for the Sissili (Arnault et al., 2016b) and the Tono (Naabil et al., 2017) basins in West Africa. Reasonable results were also achieved on a daily base for the Tana river in Kenya (Kerandi et al., 2017). In an ensemble study with the fully coupled WRF-Hydro model, encompassing six catchments
in southern Germany, Rummler et al. (2018) found that simulated and observed flow exceeding percentiles on an hourly basis were in good agreement for a three months summer period in 2005. Comparison studies with respect to WRF showed slightly improved precipitation skills with WRF-Hydro for the Crati region in southern Italy (Senatore et al., 2015) and for Israel and the eastern Mediterranean (Givati et al., 2016).

WRF-Hydro provides good capability for studying the coupled land–atmospheric boundary system from catchment to conti-
nental scale regions. Although many of the recent studies focus on classic precipitation and discharge simulation performance, the ability of the fully-coupled model system to improve physical realism for water and energy budgets across compartments becomes increasingly important and is therefore of central interest in this study.



## 2.4 Model setup and calibration

### 2.4.1 Modeling chain

The study analyzes the impact of coupling hydrological processes to the regional atmospheric modeling system WRF with respect to water and energy exchange at the land surface–atmospheric boundary layer interface. Lateral flow of infiltration excess, as well as river inflow and routing are addressed by the WRF-Hydro extension. Several parameters of WRF-Hydro influence land surface water redistribution and thus the hydrographs and therefore require thorough calibration. This is achieved by employing the standalone (i.e., not coupled to WRF) configuration of WRF-Hydro (version 3) with observations as input. Of the 8 driving variables required by the model, only observed interpolated precipitation is available with adequate coverage. Thus, the remaining input variables (temperature, humidity, wind, radiation) are taken from a standalone WRF (WRF-ARW 3.7) simulation. Altogether, as outlined in Figure 2, the modeling chain encompasses the following four steps: 1) a classic

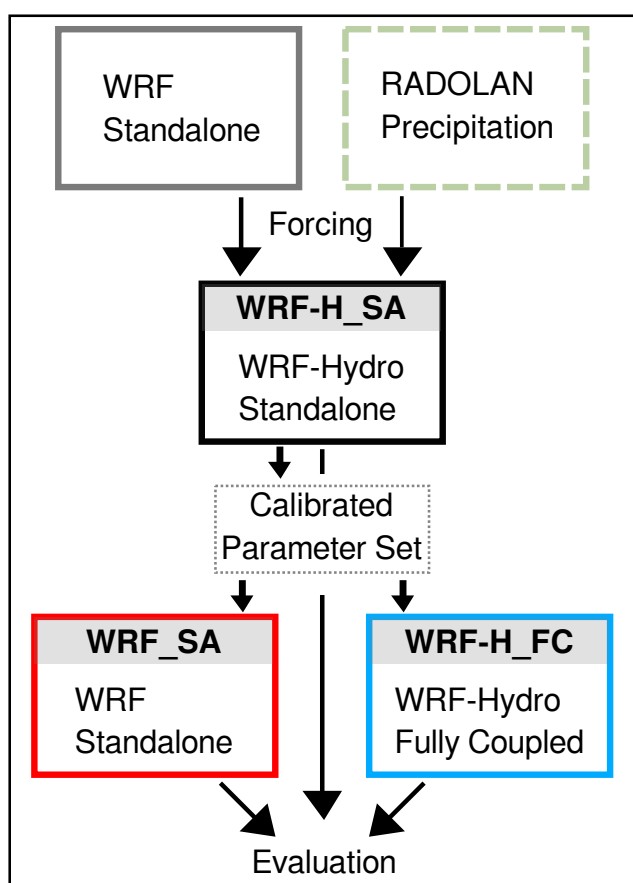

**Figure 2.** Modeling components and workflow. RADOLAN-RW is an hourly precipitation observation product of the German Weather Service with $1\,\text{km}^2$ grid resolution.





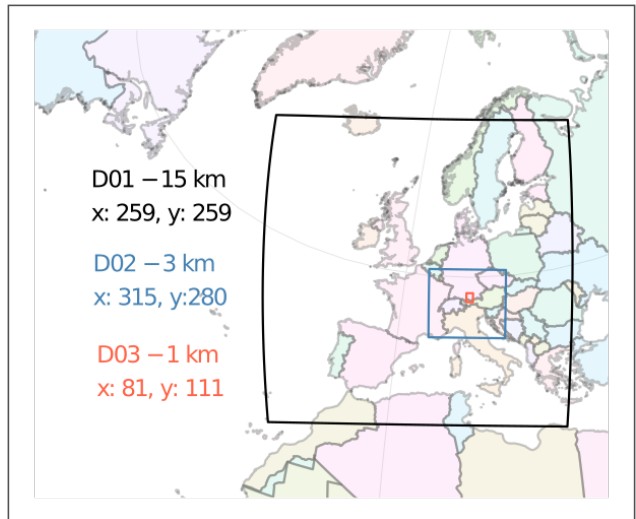

**Figure 3.** Domain nesting configuration for WRF_SA and WRF-H_FC simulations.

standalone WRF run (2015-04-01–2016-10-31) with standard LSM parameters to derive driving variables for 2) the standalone WRF-Hydro simulations (WRF-H_SA, 2015-04-01–2015-07-31 and 2016-04-15–2016-10-31) that ingest also gridded observed precipitation (RADOLAN), 3) a fully coupled WRF-Hydro simulation (WRF-H_FC 2016-04-15–2016-10-31) using calibrated parameters from WRF-H_SA, and 4) a rerun of the classic standalone WRF (WRF_SA, 2016-04-15–2016-10-31)

with the same parameter set obtained from WRF-H_SA. Finally, this leads to a commensurable set of simulations, coupled versus uncoupled.

### 2.4.2 Space and time

Figure 3 visualizes the domain and nesting configuration for the WRF_SA and WRF-H_FC simulations. A telescoping configuration with 3 nests is employed. The horizontal resolutions are 15 by 15, 3 by 3, and 1 by 1 km for domain 1, 2, and

10 3, respectively. The finest domain extends from the city of Munich in the northeast to the mountain valleys of Inn and Lech in the southwest. For all domains, the number of vertical levels is 51. Model top is defined at 10 hPa. The WRF-H_SA and WRF-H_FC simulations cover the period 2016-04-15 to 2016-10-31 including a half month for model spin-up. The starting date corresponds to snow free conditions for most of domain 3. The initial (2016-04-15) soil moisture fields for WRF_SA and WRF-H_FC are taken from the last WRF-H_SA simulation timestep (2016-10-31) to assure a 6 month spin-up period. The

15 model runs are performed continuously with none of the variables being reinitialized in between. Lateral surface water flow processes (i.e., overland and channel routing) in WRF-H_SA and WRF-H_FC are computed on a 100 by 100 m grid with the extent being identical to that of domain 3. The integration timesteps for the atmospheric part (WRF including Noah-MP LSM) are 60, 12, and 4 s for domains 1, 2, and 3, respectively. The hydrological routines are called at hourly intervals.





**Table 1.** Main WRF model physical parameterizations.

| Physics categories | Selected scheme | Reference |
| --- | --- | --- |
| Microphysics | Thompson | Thompson et al. (2008) |
| Cumulus parameterization | Kain-Fritsch* | Kain (2004) |
| Planetary boundary layer | QNSE | Sukoriansky et al. (2005) |
| Land surface model | Noah-MP | Niu et al. (2011) |
| Longwave radiation | RRTMG | Iacono et al. (2008) |
| Shortwave radiation | RRTMG | Iacono et al. (2008) |

* Only for the outermost domain

### 2.4.3 Model physics

The WRF physics parameterization for the selected domains is listed in Tab. 1. Cumulus parameterization is used only for the outer domain, while explicit convection is chosen for the finer grids, according to Skamarock et al. (2008).

For uncoupled and coupled simulations, Noah-MP (Niu et al., 2011) is used as the land surface model. The selected configu-
ration deviates from the default setup as follows: the Community Land Model (CML) method was used for stomatal resistance computation, the Schaake et al. (1996) method was used to determine infiltration and drainage (similar to classic Noah-LSM), and two stream radiation transfer applied to vegetated fraction (option 3) was applied.

The static data is adopted from the standard WRF geographic dataset. The landcover for domain one and two is based on USGS classification and includes lakes. For the innermost domain the landcover information is based on the CORINE (Büttner,
2014) dataset of the European Union, reclassified according to the USGS classes. The elevation data for the 100 m grid was derived from the ASTER global digital elevation model (ASTER GDEM, version 2). WRF-Hydro deals with several land hydrological processes not accounted for in the land surface model, such as the routing of infiltration capacity excess and saturated subsurface water. A short description of the most relevant model details for this study is provided below. Further information about technical features and standard model physics options are given in Gochis et al. (2016), while Section
2.4.4 describes the specific improvements made to the original model in order to fit with the specific features of the complex topography of this study area.

In the WRF-Hydro modeling system only subsurface and surface overland flow routing are allowed to directly affect atmo-sphere dynamics (i.e., only these processes are fully coupled). After every LSM loop, a sub-grid disaggregation loop (Gochis and Chen, 2003) is run prior to the routing of saturated subsurface and surface water, in order to achieve the desired spatial
refinement (from 1 km to 100 m) for the two state variables infiltration excess, and soil moisture content. At this stage, linear sub-grid weighting factors are assigned for preserving the sub-grid soil moisture and infiltration excess spatial variability struc-tures from one model timestep to the next. Then, subsurface lateral flow is calculated, using the method suggested by Wigmosta et al. (1994) and Wigmosta and Lettenmaier (1999) within the Distributed Hydrology Soil Vegetation Model (DHSVM). The





water table depth is calculated according to the depth of the top of the highest (i.e., nearest to the surface) saturated layer. Finally, overland flow routing also accounts for possible exfiltration from fully saturated grid cells and is achieved through a fully-unsteady, explicit, finite difference, diffusive wave approach similar to that of Julien et al. (1995) and Ogden (1997). In this study the steepest descent method is used with a timestep of 6 s. After the execution of the routing schemes, the fine grid

values are aggregated back to the native land surface model grid.

Concerning the one-way processes modeled in the WRF-Hydro system, in this study, channel flow and baseflow modules are used. Specifically, channel flow routing is performed through an explicit, one-dimensional, variable time-stepping diffusive wave formulation. Overland flow discharging into the stream channel occurs when the ponded water depth of specific grid cells, assigned to a predefined stream channel network, exceeds a fixed retention depth. The channel network has a trapezoidal

geometry, depending on the Strahler stream order functions. Currently no overbank flow is simulated.

Baseflow to the stream network is represented through a simple bucket model which uses an exponential equation to achieve the bucket discharge as a function of a conceptual water depth in the bucket. Several baseflow sub-basins (i.e., several conceptual buckets) can be specified within a watershed, but since an empirical equation is used, its parameters need to be estimated for each of the sub-basins. The baseflow model is linked to WRF-Hydro through the deep drainage discharge from the land

surface soil column. Estimated baseflow discharged from the bucket model is then combined with lateral inflow from overland flow and is input directly into the stream network as a part of the stream inflow. Total sub-basin baseflow flux to the stream network is equally distributed among all channel pixels within the sub-basin.

The reservoir module (storage of channel flow in lakes) is disabled for the simulations of this study. However, lake evaporation is considered by the Noah-MP land surface model.

**2.4.4  Changes with respect to the original WRF-Hydro model**

The model as applied in this study differs from the version 3 of WRF-Hydro with respect to soil layer representation and model timesteps. Large parts of the modeling domain and the considered river catchments exhibit mountainous terrain with steep slopes and a very thin soil layer. Here, the assumption of two meter soil thickness in Noah-LSM is not correct, as it will lead to overestimated retention of infiltrating water. Therefore, the soil features representation of Noah-LSM was changed from

two to three dimensional and soil layer depths were reduced from classic (0.1, 0.3, 0.6, 1 m) distribution to shallow values (0.05, 0.05, 0.1, 0.1 m), for all slopes >50 %, respectively. Furthermore, infiltration and percolation parameters were changed from domain uniform values to basin-wise distributed. Another important change was made with respect to the timestep configuration of WRF-H_SA and WRF-H_FC simulations. As pointed out in Senatore et al. (2015), differing intervals used for temporal integration of the Noah-LSM lead to inconsistent amounts for the soil water fluxes. The issue is caused by numeric

effects, when timesteps become very small (1 km WRF requires about 4 s) in fully coupled WRF-Hydro simulations with high resolutions. To eliminate the problem and to make all simulations comparable, the hydrology part (subgrid) in WRF-H_FC is only called at an hourly timestep, similar to that of WRF-H_SA. In WRF-H_FC the flux variables of domain 3 are therefore cumulated in between the (hourly) calls of the hydrological routines and on the other hand, overland routing output (surface head) is returned to domain 3, equally distributed over the LSM timesteps (4 s).



### 2.4.5 Driving data

Atmospheric boundary conditions for the outer domain of the WRF_SA and the WRF-H_FC simulations are derived from the ERA Interim reanalysis (Dee et al., 2011) with 0.75 ° horizontal grid spacing, 37 pressure levels from 1000 to 1 hPa, and 6 hours temporal resolution. Forcing data for WRF-H_SA are taken from the standard WRF simulation output for domain 3. The

variables comprise near surface air temperature, humidity, wind, surface pressure, short- and longwave downward radiation. Since precipitation from WRF simulation is typically biased and dislocated, an observational product of the German weather service (RADOLAN, Bartels et al., 2004; Winterrath et al., 2012) is used for substitution. It combines gauge and rain radar information and is available with an hourly timestep and $1\,km^2$ resolution.

### 2.4.6 Calibration

Different approaches for the calibration of the WRF_SA model have been followed in previous works, all based on the comparison of the observed hydrographs. Yucel et al. (2015) adopted a stepwise approach, where the parameters controlling the total water volume were first calibrated (namely, the infiltration factor, REFKDT, and the surface retention depth, RETDEPRT), and then the parameters controlling the hydrograph shape (namely, the surface roughness, OVROUGHRT, and the channel Manning roughness, MANN). Li et al. (2017), Naabil et al. (2017), Kerandi et al. (2017) and Senatore et al. (2015) followed a

similar approach. Specifically, the latter added in the first calibration step the parameter that governs deep drainage (SLOPE) and in the second step the saturated soil lateral conductivity and the bucket outflow exponent (EXPON). Furthermore, to refine calibration they introduced an automated procedure based on the *Parameter Estimation and Uncertainty Analysis software* (PEST, Doherty, 1994). Arnault et al. (2016b) mainly focused on the REFKDT and, secondarily, on the MANN parameters. Finally, Silver et al. (2017) proposed a satellite-based approach for arid (bare soil) regions aimed at calibrating topographic

slope, saturated hydraulic conductivity and infiltration parameters, based on physical soil properties and not depending on observed runoff.

Calibrating a complex hydrological model with a large number of parameters by means of only river discharge can be very problematic, particularly because of the known problem of equifinality (Hornberger and Spear, 1981; Beven, 2006; Beven and Binley, 2014). Several approaches are adopted to reduce or control this problem, particularly challenging for the emerging

fully distributed paradigm in hydrology (Beven and Binley, 1992; Beven, 2001; Kelleher et al., 2017), either constraining the parameter set by means of various strategies (e.g., Cervarolo et al., 2010) and/or incorporating different observations than discharge in the calibration process (e.g., Thyer et al., 2004; Graeff et al., 2012; Corbari and Mancini, 2014; Soltani et al., 2019). A fully-coupled atmospheric–hydrological approach further increases the degrees of freedom of the model, making the issue even more complex. In this study, while the calibration of the hydrological model is performed offline, accounting only for

discharges from several cross river sections, the effect of the resulting parameter set is evaluated considering soil, surface (both in terms of vegetation and hydrology) and atmosphere compartments all together with their reciprocal interactions. Further research will focus on more thorough analysis of equifinality issues in two-way coupled hydrometeorological models.




**Table 2.** List of parameters used for WRF-H_SA calibration.

| Parameter | Abbreviation | Sensitivity | Typical Range | Compartment |
|---|---|---|---|---|
| Surface roughness | ovrgh | yes | 0-70 mm | overland routing |
| Retention depth | retdp | yes | 0-5 mm | overland routing |
| Infiltration coefficient | refkdt | yes | 0.1-10 | LSM |
| Free drainage coefficient | slope | yes | 0-1 | LSM |
| Bucket storage height | zmax | yes | 1-500 mm | baseflow model |
| Bucket storage initial water | zinit | no | 1-500 mm | baseflow model |
| Bucket outflow coefficient | coeff | yes | 0.01-10 | baseflow model |
| Bucket outflow exponent | expon | yes | 0.1-10 | baseflow model |

After several preliminary runs, where model sensitivity to all the parameters involved in literature calibration procedures is tested, the WRF-H_SA model calibration followed also a two step approach, but in a different sense with respect to Yucel et al. (2015). First, the *Latin-Hypercube One-factor-At-a-Time* (LH-OAT, Van Griensven et al., 2006) method is used to determine the sensitivity of a set of 8 selected parameters on sub-basin river discharge, but also to obtain a starting configuration for

5   automatic parameter optimization. In a second step, 7 sensitive parameters are optimized for the 6 different sub-basin outlets (Fig. 1b) using PEST (Doherty, 1994). Table 2 gives an overview of the parameters and their relevance.

The calibration procedure is adopted for the different subcatchments in cascade, starting from upstream (i.e., parameters are first calibrated for Am-OAG, Ach-OBN and Rt-RST, then for Am-PEI and Ach-OBH and, finally, for Am-WM, see Fig. 1b). For LH_OAT the goodness of fit is determined using the volumetric efficiency (VE, Criss and Winston, 2008):

$$VE = 1 - \frac{\sum |Q_{obs} - Q_{sim}|}{\sum Q_{obs}}$$

10   with $Q_{Obs}$ and $Q_{sim}$ denoting the observed and simulated discharge in $m^3 s^{-1}$, and the Nash-Sutcliffe Efficiency (NSE). PEST optimization relies on an objective function given by the sum of squared deviations between model-generated stream-flow and observations. Table 3 lists the subcatchment-wise calibrated parameters. The LSM surface runoff scaling parameter REFKDT is globally set to $2e^{-06}$ as smaller values would have decreased infiltration to very small amounts.

Since the study focuses on land atmosphere exchange, and river routing has no feedback to the LSM, the channel parameters 15   (geometry, roughness coefficient) are not further optimized with respect to peak timing.

The calibration period length of 3.5 months is selected as a compromise between the number of model runs (about 2000), required during hypercube sampling and PEST optimization, and the available computational resources.

The hydrographs of the calibrated WRF-H_SA simulation are presented in Figure 4. The final parameter sets and goodness of fit measures are listed in Table 3. For all subcatchments, reasonable configurations could be determined. The three upper 20   Ammer subcatchments (OAG, PEI, WM) required adding a constant baseflow contribution of 2, 4.71, and 5.13 m³ s⁻¹ to the





**Table 3.** Calibration parameters for the six subcatchments of Ammer (Am), Ach, and Rott (Rt), at the gauges Oberammergau (OAG), Peißenberg (PEI), Weilheim (WM), Obernach (OBN), Oberhausen (OBH), and Raisting (RST)

| Basin | Am-OAG | Am-PEI | Am-WM | Ach-OBN | Ach-OBH | Rt-RST |
|---|---|---|---|---|---|---|
| coeff | 5.00 | 11.00 | 2.41 | 3.10 | 4.27 | 0.010 |
| expon | 3.00 | 11.00 | 0.01 | 1.50 | 1.58 | 0.631 |
| zmax | 1.00 | 1.41 | 58 | 1.00 | 29 | 1.0 |
| refkdt | 0.74 | 0.028 | 0.196 | 0.061 | 3.73 | 0.106 |
| slope | 0.25 | 0.20 | 0.14 | 0.05 | 0.06 | 0.025 |
| ovrgh | 65.2 | 15.60 | 37.6 | 40.9 | 5.88 | 26.30 |
| retdp | 0.73 | 5.00 | 0.64 | 3.16 | 3.82 | 0.100 |
| VE | 0.54 | 0.56 | 0.50 | 0.54 | 0.78 | 0.46 |
| NSE | 0.36 | 0.21 | 0.68 | 0.56 | 0.60 | 0.56 |
| Shift | 2.00 | 4.71 | 5.13 | - | - | - |
| VE_shifted | 0.81 | 0.72 | 0.79 | - | - | - |
| NSE_shifted | 0.64 | 0.64 | 0.62 | - | - | - |

model output, respectively. Channel inflow from the valley bottom or deeper large aquifers is not taken into account by WRF-Hydro's conceptual bucket scheme.

The calibration performed in the spring/summer of 2015 is validated over the period 2016-05-01–2016-10-31 (Fig. 5). Performance statistics are in general comparable, except for the reduced NSEs for Ach-OBH, Am-PEI and Am-WM. One

reason for the decrease in performance is the simulated but not observed discharge peak on 2016-06-29 which is caused by an erroneous precipitation observation in RADOLAN where there is no rainfall in the region on that day at all. Furthermore, in the case of Ach-OBH, as expected due to the disabling of the reservoir option, the buffering effect of the lake cannot be reproduced by the model thus leading to an overestimation of the peaks. Finally, Am-WM aggregates the mismatches of all the upstream subcatchments.

The catchment-based, lumped calibration of hydrological parameters in WRF-Hydro seems to be rather limited. Especially, for complex terrain as presented by this study, the distribution of discharge gauges does not agree with landscape units. Therefore, the lumped parameter sets have to union quite diverse subcatchment conditions which may lead to unrealistic spatial representations of the physical properties they represent. Thus, for further studies, it is recommended to find parameter sets that are bound to landscape characteristic, such as relief, landcover type and soil features (e.g., Hundecha and Bárdossy, 2004;

Samaniego et al., 2010; Rakovec et al., 2016; Silver et al., 2017), which also contribute to reduce the equifinality problem (Kelleher et al., 2017).



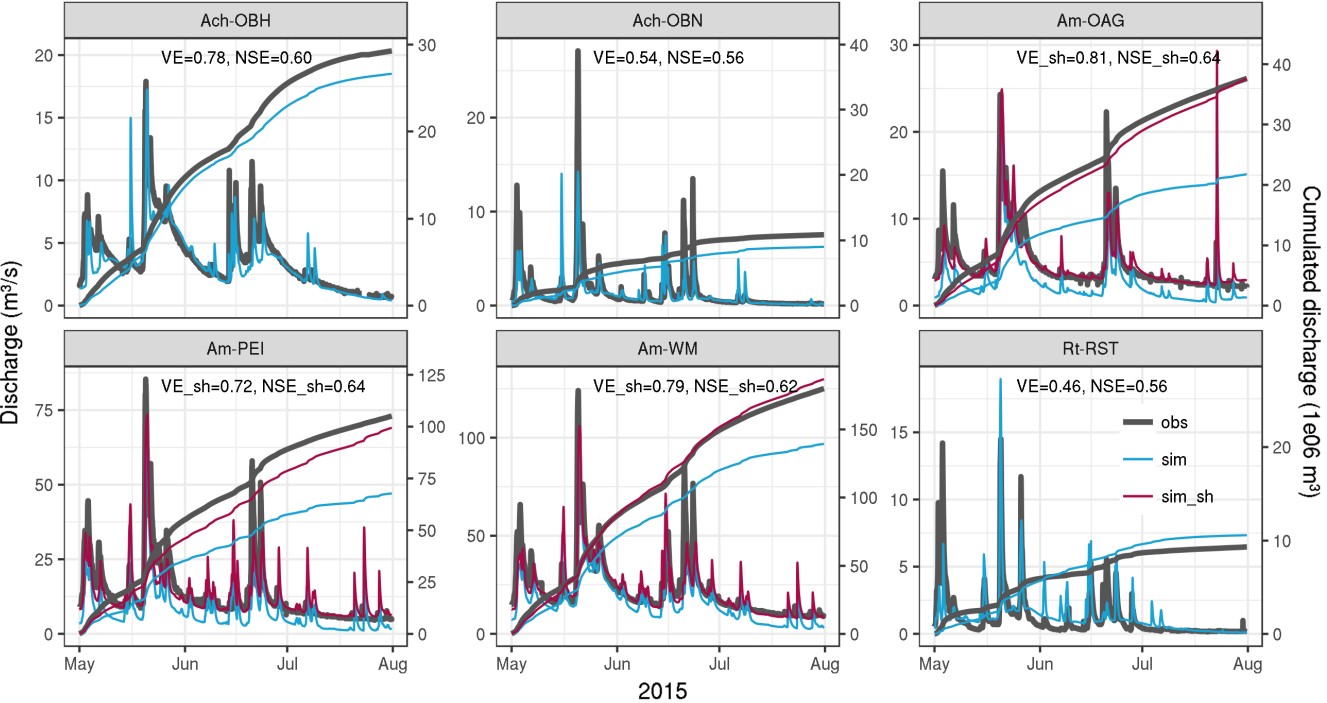

**Figure 4.** Subcatchment hydrographs for calibration period. Standard WRF-H_SA model output is printed in blue. Shifted (sh) hydrographs are shown in red. Shift amounts are listed in Tab. 3.

## 3 Results and discussion

The following section evaluates and discusses the simulations of the standalone WRF (WRF_SA) and the fully coupled WRF-Hydro (WRF-H_FC). In the first part, based on the TERENO-preAlpine stations, the energy fluxes at the land–atmosphere boundary are analyzed, in particular radiation, heat fluxes, and near surface air temperature, evapotranspiration and soil mois-

5 ture. The second part compares modeled and observed atmospheric boundary layer profiles for the DE-Fen site. The third part deals with the subcatchment aggregated water budgets and looks at the differences in the temporal evolution of simulated soil moisture patterns.

### 3.1 Model evaluation for TERENO-preAlpine stations

#### 3.1.1 Radiation

10 The evaluation of WRF_SA and WRF-H_FC simulations with observations from TERENO-preAlpine focuses on the radiation input, its partitioning into water and energy fluxes at the land surface and on the near surface atmospheric and subsurface states. Fig. 6 shows the mean diurnal cycles of simulated and observed downward surface shortwave radiation for three TERENO-preAlpine sites for the period June to October 2016. For all locations, the simulations overestimate radiation from sunrise to





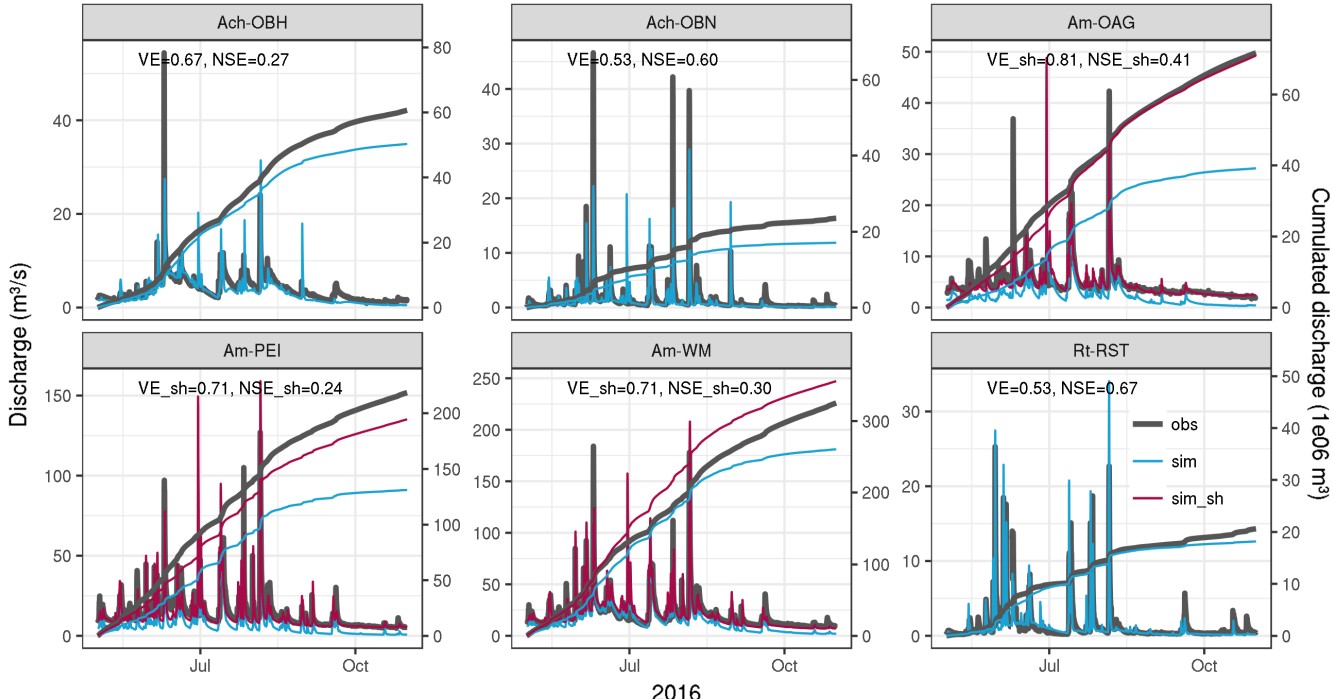

**Figure 5.** Subcatchment hydrographs for validation period (2016-05-01 – 2016-10-31). Standard WRF-H_SA model output is printed in blue. Shifted (sh) hydrographs are shown in red. Shift amounts are listed in Tab. 3.

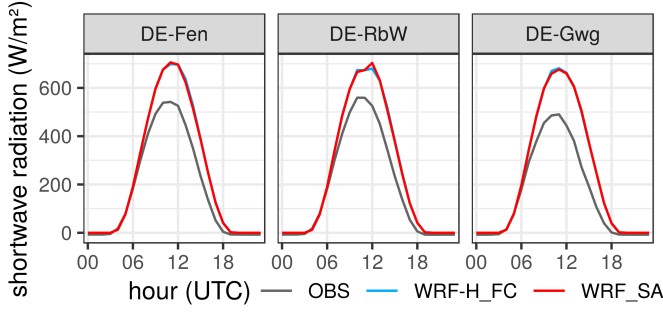

**Figure 6.** Mean diurnal cycles of simulated and observed downward surface shortwave radiation for June to October 2016.

sunset with similar magnitude. While correlations for the hourly values are high ($r^2$: DE-Fen 0.76, DE-RbW 0.73, DE-Gwg 0.66), the mean errors (ME W/m$^2$: DE-Fen 61, DE-RbW 57, DE-Gwg 78) reveal considerable bias. Also the root mean square errors show substantial scatter (RMSE W/m$^2$: DE-Fen 163, DE-RbW 169, DE-Gwg 194). The overestimation of summer shortwave radiation for Central Europe with WRF has also been documented by other studies and is usually related to under-

5  estimated cloud cover, especially in the mid troposphere where convection is active (García-Díez et al., 2015; Katragkou et al.,





2015). The increased bias for DE-Gwg could be related to local shading due to topography in this narrow Alpine valley and because of higher convective activity in this mountain region. The comparison of WRF_SA and WRF-H_FC simulation does not yield considerable differences.

The results for downward longwave radiation are given in Fig. 7. The negative biases for the different locations (ME W/m$^2$:

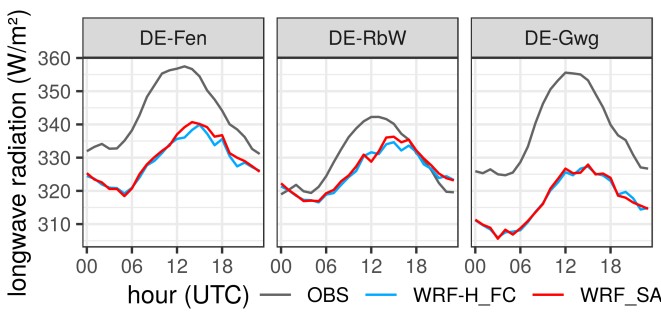

**Figure 7.** Mean diurnal cycles of simulated and observed downward longwave radiation for June to October 2016.

DE-Fen -14, DE-RbW -4, DE-Gwg -21) correspond with the above suggested cloud cover underestimation. With -1 to -6 %, the relative deviations are rather small. Again, the differences between standalone and coupled model are nominal. Similar overestimation is obtained for total absorbed shortwave radiation (Tab. S1). Thus, for the TERENO-preAlpine locations it can be stated that shortwave radiation input to the land surface is overestimated with 33 to 56 % by both standalone and coupled model.

**3.1.2  Heat fluxes**

The diurnal cycles of latent and sensible heat fluxes are presented in Figs. 8 and 9 for three different grassland sites. The analysis is split by month from June to October 2016, to provide insight about the temporal variations. Observation data for DE-RbW is missing from 25$^{th}$ September to end of October. The coupled WRF-H_FC simulation exhibits increased latent and decreased sensible heat fluxes for the DE-Fen and DE-RbW sites, whereas WRF_SA and WRF-H_FC are very similar

for the Alpine valley location (DE-Gwg). In the case of sensible heat flux, the hydrologically enhanced model (WRF-H_FC) outperforms or is equal to the standalone simulation (WRF_SA). For latent heat the mean diurnal fluxes are overestimated for DE-Fen for June to August by either both models or the coupled run. A constant positive bias is also found for DE-Gwg (except for Oct.). Tabs. 4 and 5 list the performance measures for latent and sensible heat for the period June to October 2016. For latent heat, correlation improves considerably for DE-Fen and DE-RbW with the coupled model but ME and RMSE deteriorate

for DE-Fen and DE-Gwg. Overall improvement for WRF-H_FC with respect to the observations is only obtained for DE-RbW. For sensible heat, the coupled run yields improved performance for DE-Fen and DE-RbW and is also in good agreement with the observations at DE-Gwg. Ground heat flux (Fig. S1) is overestimated by the models from about two hours after sunrise until noon. From afternoon till dawn, both simulations overestimate the upward (land to atmosphere) radiative flux. WRF_SA





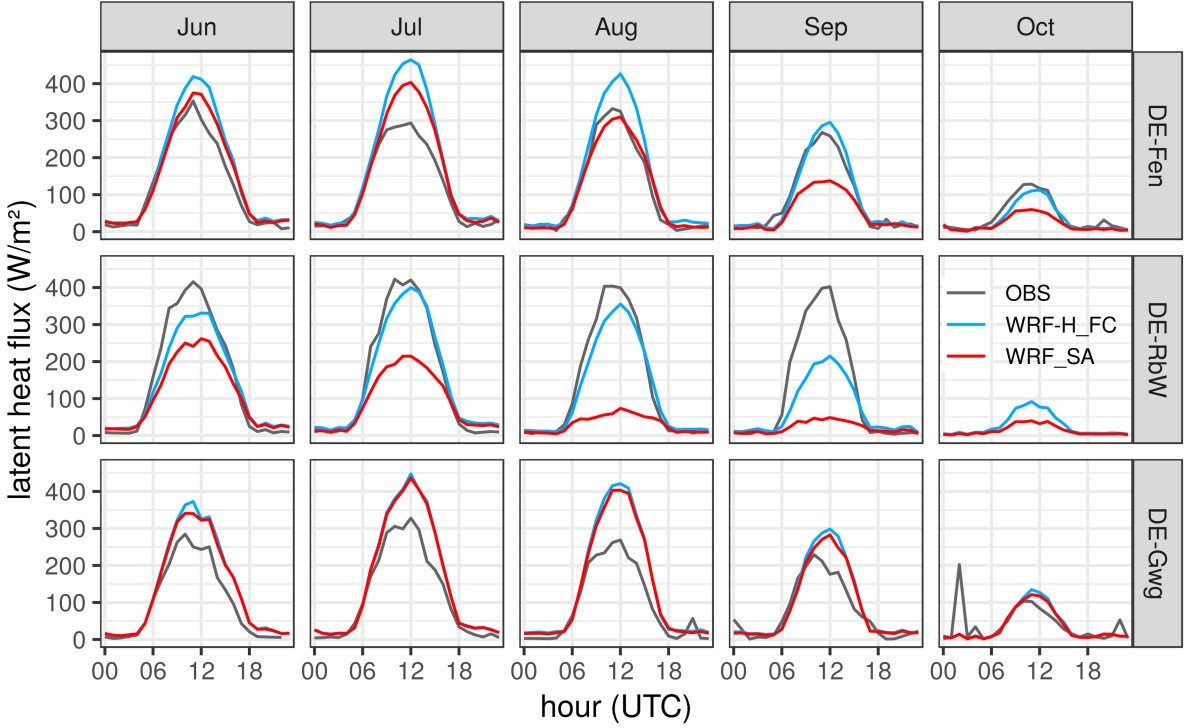

**Figure 8.** Mean diurnal cycles of simulated and observed latent heat flux for the months June to October 2016 at different TERENO-preAlpine sites.

**Table 4.** Performance measures for latent heat flux (W m$^{-2}$), simulations vs. observations, June to October 2016.

| Station | Model | $r^2$ | ME | RMSE |
|---------|-------|-------|------|------|
| DE-Fen | WRF_SA | 0.59 | 3.96 | 93.77 |
| | WRF-H_FC | 0.67 | 38.33 | 100.00 |
| DE-RbW | WRF_SA | 0.44 | −73.58 | 150.09 |
| | WRF-H_FC | 0.77 | −6.59 | 82.99 |
| DE-Gwg | WRF_SA | 0.60 | 48.05 | 106.95 |
| | WRF-H_FC | 0.59 | 53.42 | 112.34 |

and WRF-H_FC differ slightly for ground heat fluxes, with WRF-H_FC showing slightly increased performance with respect to the observations (Tab. S2).





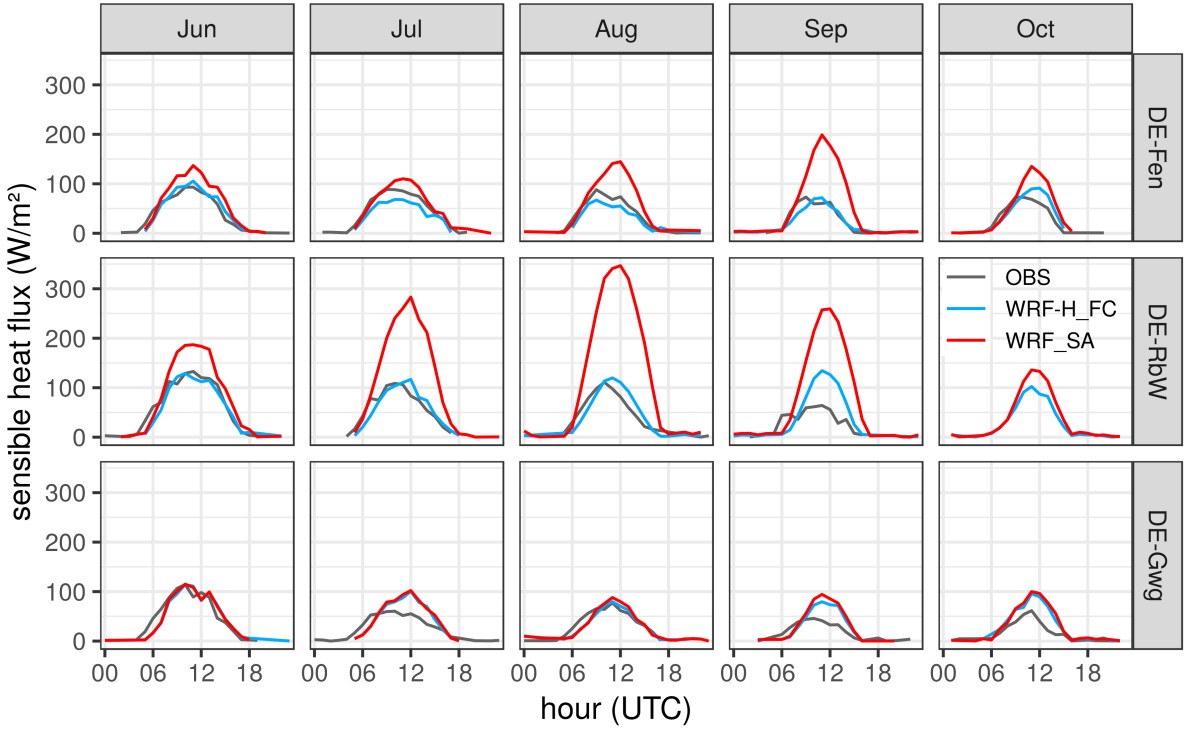

**Figure 9.** Mean diurnal cycles of simulated and observed sensible heat flux for the months June to October 2016 at different TERENO-preAlpine sites.

**Table 5.** Performance measures for sensible heat flux (W m$^{-2}$), simulations vs. observations, June to October 2016.

| Station | Model | $r^2$ | ME | RMSE |
|---|---|---|---|---|
| DE-Fen | WRF_SA | 0.37 | 17.99 | 62.83 |
| | WRF-H_FC | 0.41 | −6.46 | 45.52 |
| DE-RbW | WRF_SA | 0.48 | 46.43 | 104.01 |
| | WRF-H_FC | 0.59 | −6.98 | 40.86 |
| DE-Gwg | WRF_SA | 0.23 | 5.18 | 55.25 |
| | WRF-H_FC | 0.23 | 2.00 | 53.63 |

### 3.1.3 Near surface temperature and humidity

The diurnal course of 2 m air temperature (Fig. S2) is similar for uncoupled and coupled model for June, October for all stations. The mountain peak stations (Kol, LaS) and the alpine valley station (DE-Gwg) are hardly sensitive to coupling.





Prominent deviations between WRF_SA and WRF-H_FC occur for July to September at the foreland stations (DE-Fen, DE-RbW, Ber). Here, the coupled simulations between 06 and 18 UTC agree better with the observations. Nighttime values are generally overestimated by both models and coupling does not have an influence. The mean errors improve between 0.34 and 0.6 K for the foreland and between 0.11 and 0.25 K for the mountain stations whereas correlation remains identical. Slight

5   improvement with coupling is also obtained for the RMSE values (Tab. S3).

Figure 10 provides the monthly diurnal cycles for 2 m mixing ratio. The model comparison reveals higher values for the coupled model run, especially during sunshine hours (06-18 UTC). Also prominent is a peak in 2 m moisture around 1700 UTC for both models that is not as pronounced in the observations. For July to August, the coupled simulation resembles

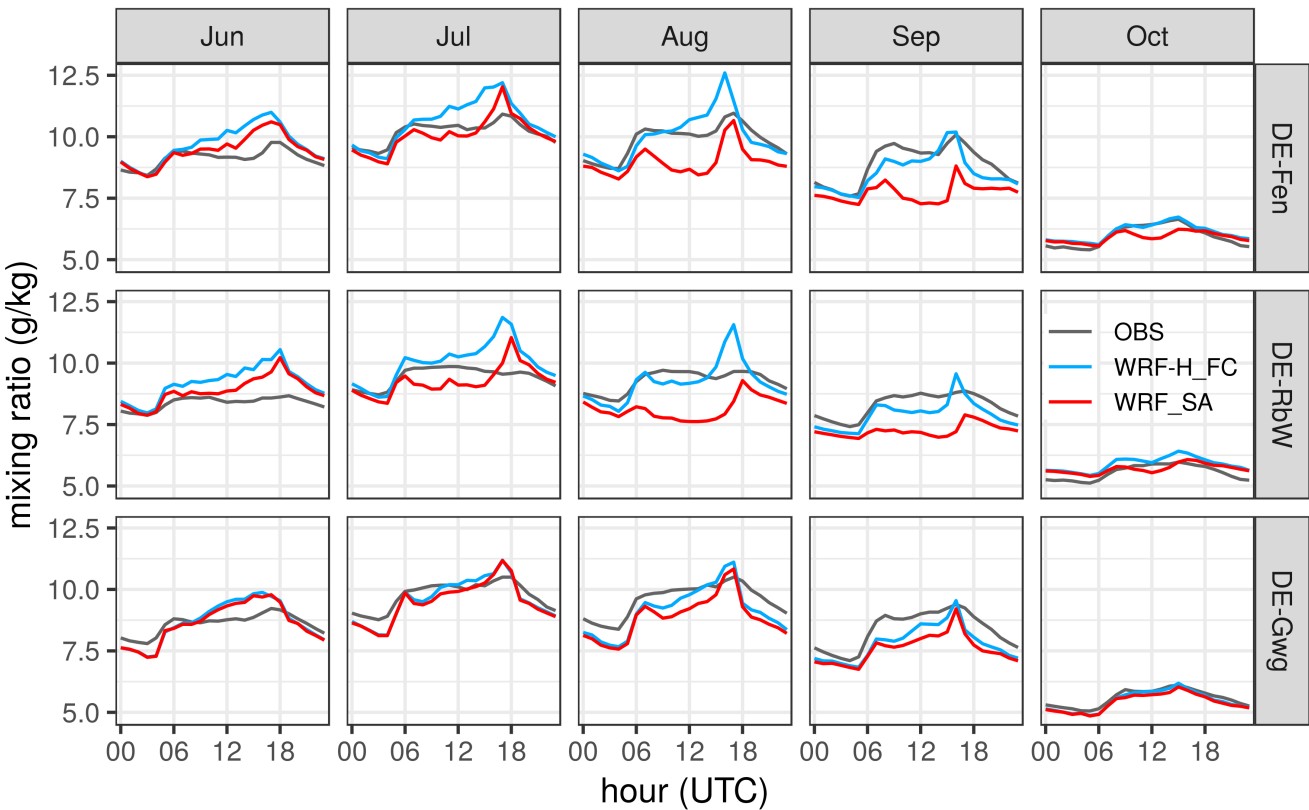

**Figure 10.** Mean diurnal cycles of simulated and observed 2 m mixing ratio for the months June to October 2016 at different TERENO-preAlpine sites.

the observations better for the morning rise in moisture concentration but towards the afternoon, the constant rise exceeds the

10   measurements. According to the performance measures in Table 6, the correlation increases considerably with the WRF-H_FC configuration for the DE-Fen and DE-RbW sites. Also ME and RMSE are reduced. For DE-Gwg the findings are likewise, however the magnitudes are lower.





**Table 6.** Performance measures for 2 m mixing ratio (g kg$^{-1}$), simulations vs. observations, June to October 2016.

| Station | Model | $r^2$ | ME | RMSE |
|---------|-------|------|------|------|
| DE-Fen | WRF_SA | 0.57 | −0.38 | 1.60 |
|        | WRF-H_FC | 0.69 | 0.19 | 1.35 |
| DE-RbW | WRF_SA | 0.46 | −0.35 | 1.66 |
|        | WRF-H_FC | 0.62 | 0.25 | 1.42 |
| DE-Gwg | WRF_SA | 0.67 | −0.39 | 1.34 |
|        | WRF-H_FC | 0.70 | −0.24 | 1.26 |

Altogether, it can be stated that the hydrologically enhanced setup (i.e., WRF-H_FC) leads to an improved representation of 2 m temperature and mixing ratio.

### 3.1.4 Evapotranspiration and soil moisture

Figure 11 shows the mean diurnal cycles for simulated and lysimeter-observed evapotranspiration. The coupled simulation 5 leads to a generally higher flux, especially for July to September. As obtained for heat fluxes and mixing ratio, the deviations between WRF-H_FC and WRF_SA are small for DE-Gwg. At the other sites, the differences are up to fourfold. The lysimeter observations divide into intensive and extensive management which differ in cutting frequency, amount of slurry application, and plant species composition. The extensive management type corresponds with the land use in the footprint of eddy-covariance station. However, the discrepancies between the two scenarios are usually not very pronounced. For June to 10 July, the simulations tend to exceed the observations for DE-Fen whereas in the same period, at DE-RbW the coupled model agrees well with the lysimeters whereas WRF_SA considerably underestimates. For the June to October period WRF-H_FC shows improved performance in terms of $r^2$ (Tab. 7). The relative bias ranges between 43 and -48% and the directions of deviation agree with the findings for the latent heat evaluation at the flux towers.

Observed and simulated soil moisture for the DE-Fen site are presented in Figure12. The gray ribbons depict the 25-75 15 percentiles of a wireless soil moisture sensor network that consists of 55 profiles with measurements at 5, 20, and 50 cm depth (SoilNet, further details are available in Kiese et al., 2018; Fersch et al., 2018). Obviously, simulations and observation show a considerable offset and also the temporal variations are much smoother for the model. The discrepancies in the soil moisture time-series are largely attributable to the difference in saturation water content. The LSM assumes loam for the DE-Fen site (and almost for the entire area of domain 3) with a maximum volumetric soil moisture content of 44 % whereas in reality the 20 region consists of sandy to silty loams and also peaty areas where the maximum soil moisture ranges between 50-80 %. With WRF-H_FC the soil moisture values are about 8-10 % higher than with WRF_SA. Also the decline differs between the two with WRF_SA leading to a much dryer scenario for the summer months. That is where WRF-H_FC-simulated latent heat flux





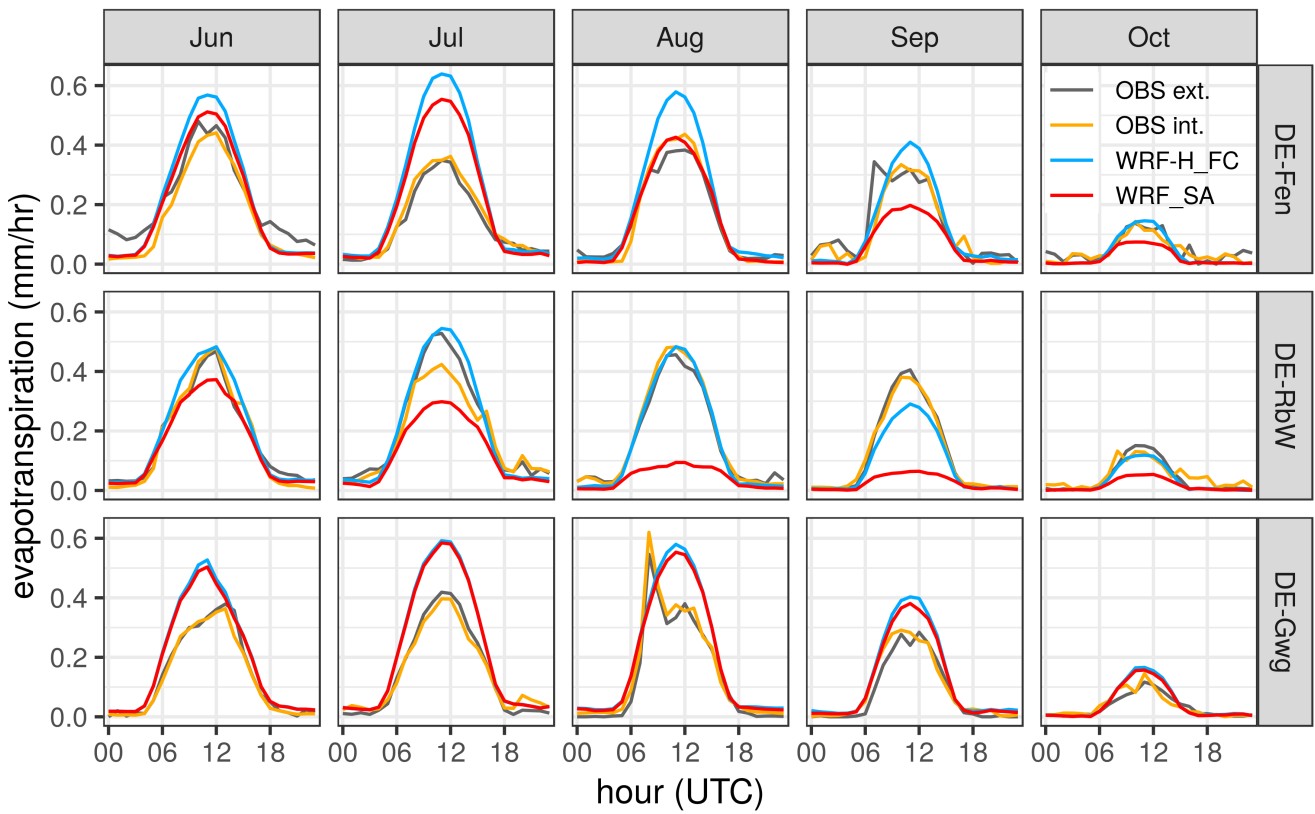

**Figure 11.** Mean diurnal cycles of simulated and observed grassland evapotranspiration for the months June to October 2016. Black and yellow represent the lysimeter derived ET for extensive and intensive land-use scenarios.

**Table 7.** Performance measures for evapotranspiration (mm hr$^{-1}$), June to October 2016.

| Station | Model | $r^2$ | ME | RMSE |
|---------|-------|-------|------|------|
| DE-Fen | WRF_SA | 0.32 | 0.00 | 0.19 |
| | WRF-H_FC | 0.34 | 0.04 | 0.20 |
| DE-RbW | WRF_SA | 0.25 | −0.07 | 0.21 |
| | WRF-H_FC | 0.44 | 0.00 | 0.17 |
| DE-Gwg | WRF_SA | 0.26 | 0.04 | 0.24 |
| | WRF-H_FC | 0.25 | 0.05 | 0.24 |





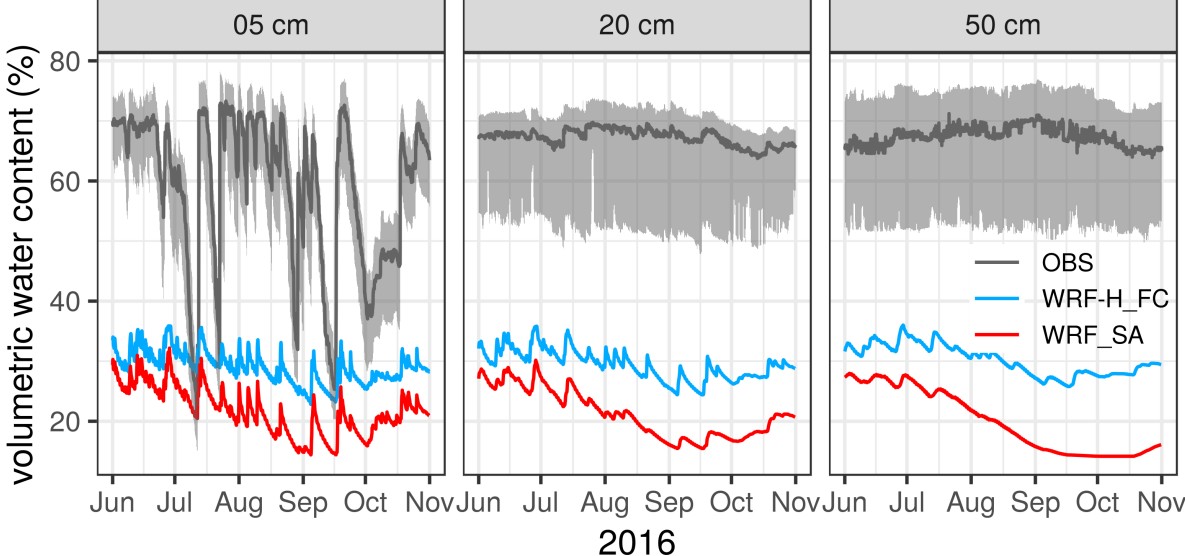

**Figure 12.** Volumetric water content at DE-Fen site for three different depths. Gray ribbon: range between first and third quartile of SoilNet observations; gray line: median of SoilNet observations

and evapotranspiration largely outperform. Altogether, for DE-Fen, the decline predicted by WRF-H_FC seems more realistic with respect to the observations. This is also confirmed by the mostly improved statistical measures (Tab. 8). The representation of soil moisture in LSMs is a general challenge. Soil parameters and water contents are often tuned to unrealistic values for the sake of obtaining a good matching of the surface exchange fluxes with observations (Koster et al., 2009). The recent

publications of global and continental high resolution soil hydraulic datasets (like, e.g., Hengl et al., 2017; Tóth et al., 2017) are helpful to improve and unify soil moisture representations in those models. However, these datasets, with their underlying water retention models (e.g., Van Genuchten, 1980), are not supported by the Noah LSMs and an implementation would be out of the scope of this study.

### 3.2   Boundary layer profiles during ScaleX campaign 2016

Figure 13 shows the spline interpolated vertical profiles of the performance measures for the WRF_SA and WRF-H_FC simulations and the HATPRO observations for the planetary boundary layer. The measurements represent hourly subsampled time-series from 2016-06-01 to 2016-07-31 for air temperature (Fig. 13 a, b, c) and absolute humidity (Fig. 13 d, e, f). For temperature, the differences between HATPRO and the models are generally much larger than the HATPRO accuracy. For humidity, the mean error and accuracy are about the same. It is extremely likely that both models overestimate temperature and

probably underestimate absolute humidity. For the inter model comparison, the WRF-H_FC run shows reduced deviations for both variables.





**Table 8.** Soil moisture performance measures (vol. %), DE-Fen, June-October 2016.

| Depth | Model | $r^2$ | ME | RMSE |
|-------|-------|-------|-------|-------|
| 05 cm | WRF_SA | 0.28 | −37.22 | 38.83 |
|       | WRF-H_FC | 0.29 | −30.01 | 32.10 |
| 20 cm | WRF_SA | 0.07 | −46.03 | 46.20 |
|       | WRF-H_FC | 0.11 | −37.88 | 37.96 |
| 50 cm | WRF_SA | 0.01 | −47.41 | 47.74 |
|       | WRF-H_FC | 0.02 | −37.19 | 37.32 |

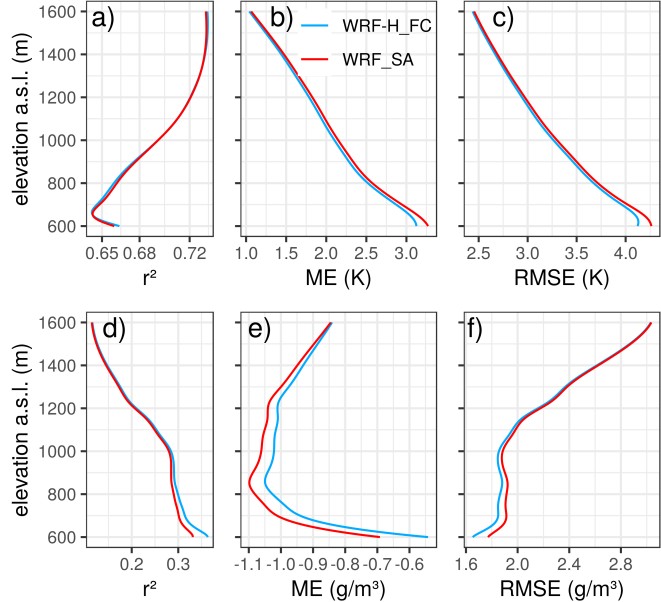

**Figure 13.** Performance evaluation of vertical atmospheric profiles of air temperature (upper panel) and absolute humidity (lower panel) for the DE-Fen site, 2016-06-01 to 2016-07-31. $r^2$ = coeffcient of determination, ME = mean error, RMSE = root mean square error.

The increase in correlation and the decrease of errors with height is only visible for temperature. The comparison between WRF_SA and WRF-H_FC reveals some modifications for the near surface region, where the fully coupled model gives a slightly improved skill. For absolute humidity, the lower parts of the profiles are in better agreement with the observations. With 0.15 to 0.35, the coefficients of determination are small as compared to 0.64 to 0.73 for temperature. The WRF-H_FC run outperforms WRF_SA especially for the lower 400 m of the profile. The time series of simulated and observed integrated water vapor for the DE-Fen site are shown in Figure 14. The temporal evolution is reasonably covered, with a few larger mismatches

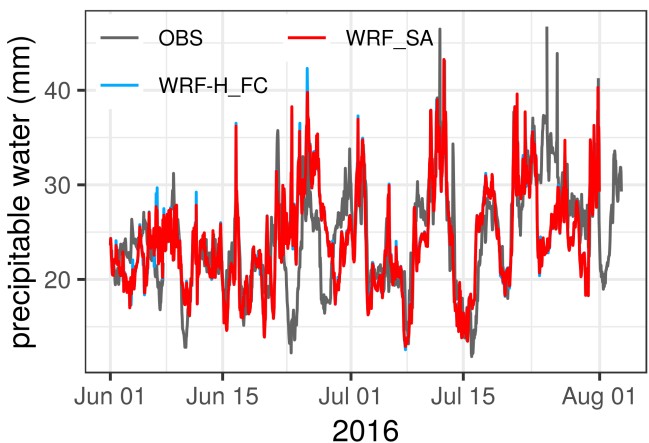

**Figure 14.** Simulated and observed precipitable water for the DE-Fen site, 2016-06-01 to 2016-07-31.

at the end of June and the end of August. The inter-model differences are very small. Both simulations show nearly identical performance with $r^2$=0.42, ME=-0.21/-0.19 mm, and RMSE=4.51/4.52 mm for WRF_SA and WRF-H_FC, respectively. The results for integrated water vapor and humidity profiles indicate that the coupling mainly affects the atmospheric boundary layer as differences in the correlation and errors between both simulations are restricted to the lower heights. Moreover, the

5 domain area seems too small for internal moisture recycling and additional precipitation generation to take place. Most of the surplus in humidity is probably transported beyond the domain boundary. If the coupled simulation was extended to a larger area, e.g., Europe, impact above the boundary layer would be expected, at least for weak synoptic forcing periods (Arnault et al., 2018).

### 3.3 Water budgets

**3.3.1 Analysis for sub-basin integrated water balances and discharge**

Figure 15 visualizes the monthly water budgets for the six different subcatchments for WRF_SA, WRF-H_FC, and WRF-H_SA (stand alone WRF-Hydro) simulations, according to the water balance equation

$$P = E + R_{SF} + R_{UG} + \Delta S_{soil}$$

with precipitation $P$, evapotranspiration $E$, surface and subsurface runoff $R_{SF}$ and $R_{UG}$, and soil storage variation $\Delta S_{soil}$. Deviations in subcatchment aggregated precipitation are small for WRF_SA and WRF-H_FC. $P$ in WRF-H_SA, that originates from the gridded observation product (RADOLAN), differs for most of the months and regions. WRF_SA and WRF-H_FC overestimate $P$ for the mountainous regions (Am-OAG and Am-PEI) for May and June. For the other subcatchments June to

15 August are mainly underestimated. September and October are well resembled for (Rt-RST). For the others, October sums are overestimated. For evapotranspiration $E$ an increasing tendency can be seen from WRF_SA via WRF-H_FC to WRF-H_SA.

**Figure 15.** Monthly water budgets in mm for subcatchments Ach, Ammer (Am), and Rott (Rt). P = precipitation, E = evapotranspiration, $\Delta S_{soil}$ = soil water storage change, $R_{sf}$ = surface runoff, $R_{UG}$ = underground runoff (groundwater recharge). WRF_SA = standalone WRF model, WRF-H_FC = fully-coupled WRF-Hydro model, WRF-H_SA = observation driven WRF-Hydro standalone model.





Underestimated $P$ in WRF_SA and WRF-H_FC results in decreased $E$. For the Am-OAG subcatchment, the overestimation in $P$ is transferred to surface and subsurface runoff. This is likely because of the soil moisture in the mountain region is generally higher and $E$ is rather energy limited and therefore cannot increase considerably. Moreover, slopes are steeper and soil thickness is reduced so that percolation takes place quickly. The variation in soil volumetric water content is irregular

among models and for the different months. This indicates a non-linear feedback for the land–atmosphere interaction. Soil infiltration generally increases with the hydrologically enhanced models, however, the amounts for WRF-H_FC and WRF-H_SA vary according to $P$. Storage depletion (negative values) does not exhibit any tendency among the different models. Surface runoff (infiltration excess) with WRF_SA is 50%, in some cases more than 100%, higher than with WRF-H_FC and WRF-H_SA. Conversely, groundwater recharge (soil drainage) increases for the hydrologically enhanced models. Again,

differences between WRF-H_FC and WRF-H_SA are due to the individual precipitation amounts. On monthly scale, changes of the canopy water storage compensate (not shown). The water budget residuals, caused by the subgrid aggregation and disaggregation and by other numerical artifacts, can reach up to 31 mm for WRF-H_FC at Rt-RST in September but in the mean they are 5.6 mm. Altogether, the coupling with hydrology leads to increased infiltration, slightly increased $E$ but almost no changes in $P$. A reason for this could be that the distance of the displacement between precipitation generation and falling

locations is generally much larger than the one covered by the domain boundaries in this study (Arnault et al., 2016a; Wei et al., 2015).

Table 9 lists the performance measures for the discharge simulated with the fully coupled model (hydrographs available in the supplementary material Fig. S3) with baseflow shifted simulations (compare Tab. 3, WRF-H_SA calibration) denoted by *sh*. Nash-Sutcliffe efficiencies are poor for all stations, Kling-Gupta efficiencies (KGE', Kling et al., 2012) lie between 0.07

and 0.39 with a general performance gain for the baseflow shifted time-series. For all subcatchments except Am-OAG, negative MEs are observed, with values ranging from $-22.6$ to $2.98$ mm mon$^{-1}$. For the three Ammer gauges (OAG, PEI, WM), adding the baseflow shifting leads to an improved baseline of the hydrographs and volumetric efficiency but also to considerable overestimation of the cumulated sums. If the non-shifted MEs for $Q$ are compared with those of $P$ it turns out that for some of the subcatchments (Am-WM, Ach-OBH, Rt-RST), the deviations are of similar amount as precipitation bias. The poor

performance of the fully-coupled simulation to predict hourly discharge can be clearly mapped to the model's difficulty to reproduce the timing and positioning of precipitation.

### 3.3.2 Spatial variations of simulated soil moisture patterns

Figure 16 shows the time-series of root mean square deviations (RMSD) of the spatial variograms for the WRF_SA and WRF-H_FC simulations subdivided by the four soil layers in the model. The variograms were computed using ten equidistant

lags of 1 km from 1 to 10 km. The RMSDs were computed for the six different subcatchments and also for the Ach, Ammer and Rott catchments, and for the full domain. For the calculation, the subregions were masked so that the adjacent areas and lakes did not impact the results. The analysis reveals that the structural differences between the two models have their maximum in late summer and fall. Surprisingly, layer three gives the strongest variations in spatial patterns. The changes for layers one and two are not so pronounced. The cause for this might be that the thinner top layers are strongly influenced by





**Table 9.** Discharge performance measures for subcatchment gauges modeled with the fully coupled WRF-H_FC model, June to October 2016. The gauges marked with *sh* denote the shifted discharge simulation results as applied in the WRF-H_SA calibration (Tab. 3). Mean errors (ME) for Q (discharge) and (P) are given in mm mon$^{-1}$.

| Gauge | NSE | VE | KGE' | ME (Q) | ME (P) |
|---|---|---|---|---|---|
| Am-OAG | −1.61 | 0.44 | 0.14 | 2.98 | 17.20 |
| Am-OAG_sh | −2.11 | 0.37 | 0.17 | 22.32 | 17.20 |
| Am-PEI | −0.66 | 0.42 | 0.12 | −16.62 | 1.70 |
| Am-PEI_sh | −0.63 | 0.53 | 0.39 | 12.34 | 1.70 |
| Am-WM | −0.56 | 0.46 | 0.25 | −16.23 | −13.90 |
| Am-WM_sh | −0.55 | 0.52 | 0.35 | 12.43 | −13.90 |
| Ach-OBH | −0.22 | 0.44 | 0.31 | −14.03 | −12.10 |
| Ach-OBN | −0.02 | 0.26 | 0.07 | −22.60 | −13.20 |
| Rt-RST | −0.56 | 0.05 | 0.20 | −1.55 | −4.73 |

precipitation and infiltration processes and thus the spatial patterns are shaped accordingly. Due to its larger thickness, layer three reacts more sluggish. Thus deviations are more persistent. Layer four is even thicker but shows only slight variations over time. It is likely that the free drainage boundary condition has a regulatory effect here. Also the withdrawal of water for plant transpiration is partly reduced, as only forest landcover classes have roots in this layer per definition. Am-OAG depicts
a special case as soil layers are considerably thinner for the steep mountain slopes (see 2.4.4). Thus, response times are short and routing of infiltration excess water is quickly propagated through all layers. The variability for the united Ach, Ammer and Rott catchments is less pronounced than seen for the smaller entities, but still the maximums are from late summer to fall, and layer three is affected most.

## 4 Summary, conclusive remarks and perspectives

The calibration of water related land surface parameters is hardly used for local area and regional climate model applications. The incorporation of water budgets in the model optimization provides an additional means to evaluate with independent observations. Such a concept requires a coupled atmospheric–hydrological approach that relates the land surface to planetary boundary layer exchange of energy and water with the spatial redistribution processes of water thus enabling the closure of the regional water balance and complex feedback processes at the land–atmosphere boundary.

This study examines the skills of a classic and a hydrologically-enhanced–fully-coupled setup of the Weather Research and Forecasting model (WRF / WRF-Hydro) to reproduce the land–atmosphere exchange of energy and water as well as the water budgets of the Ammer and Rott watersheds in southern Germany, for a six month period of 2016. The evaluation is based





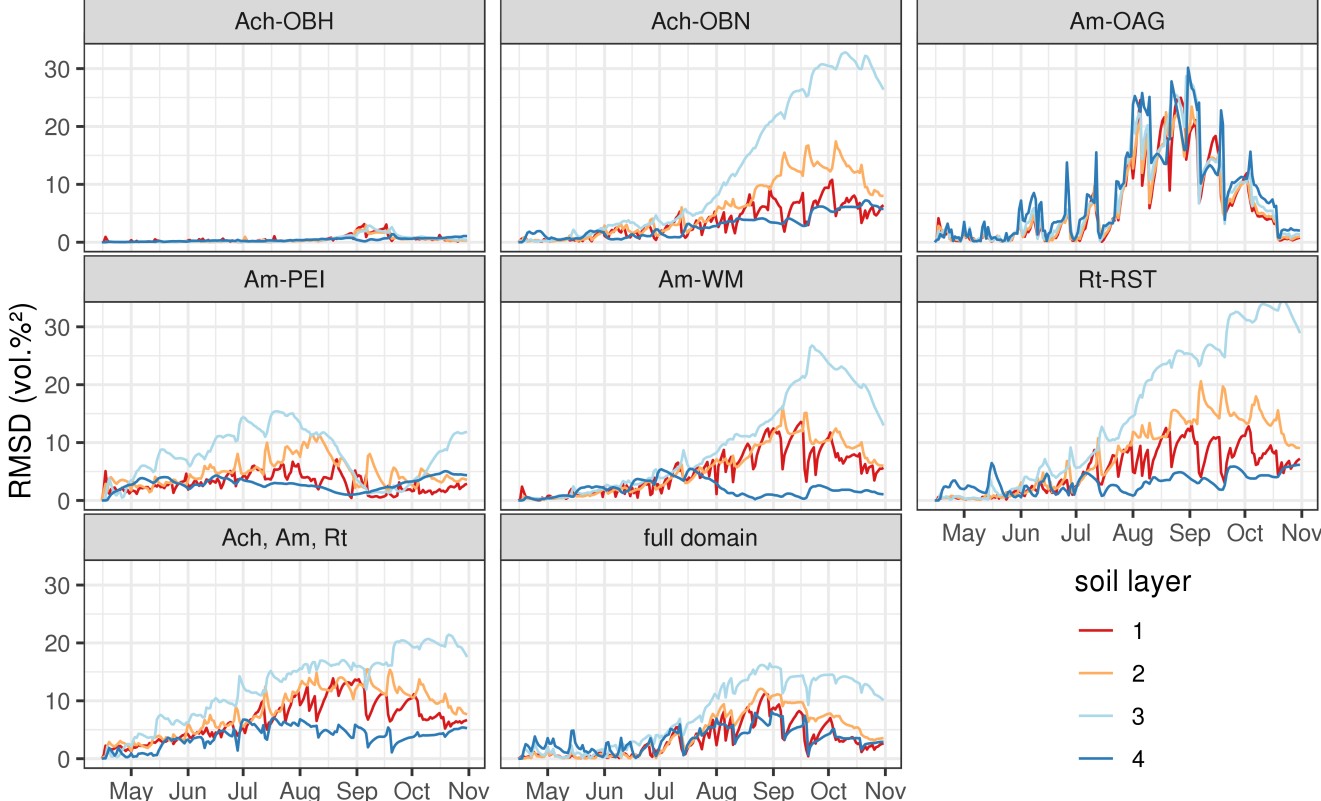

**Figure 16.** Time-series of root mean square deviations between empirical variograms of WRF_SA and WRF-H_FC for soil moisture layers 1 to 4, based on 10 equidistant lags (1 to 10 km).

on comprehensive observations available by the TERENO-preAlpine environmental observatory and further third-party data suppliers such as the Bavarian Environmental Agency.

A standalone version of the WRF-Hydro model (without the atmospheric part), driven by WRF-simulated meteorological variables and observed precipitation (DWD-RADOLAN) was calibrated for six different subcatchments and the resulting

5   parameters were subsequently used for a standalone WRF and a fully-coupled WRF-Hydro simulation both being identical with respect to initialization, parameters, forcing and binary code.

The calibration of the standalone WRF-Hydro model (WRF-H_SA) based on observed precipitation yielded reasonable results in terms of Nash-Sutcliffe and volumetric efficiencies. In some cases it was required to correct for an underestimated constant baseflow contribution. As compared to the standard settings in WRF and Noah-MP, for all subcatchments except Ach-

10   OBH, the surface infiltration parameter REFKDT needed to be reduced (associated with higher infiltration), to improve the simulated hydrographs. The volumetric efficiency measure was an important indicator for further optimizing the parameters when Nash-Sutcliffe and Kling-Gupta efficiencies already converged. Also the percolation parameter SLOPE was mainly





reduced as compared to the standard value, meaning that a considerable portion of former infiltration excess water needed to be transferred to the bucket-storage to assure good performance for the simulated baseflow. For the validation period, the skill measures deteriorated, also because of artifacts in the RADOLAN precipitation product, however still with reasonable hydrographs. It is concluded that some processes cannot be depicted by the model physics or because of the lumped parameter

estimation approach. Altogether, the subcatchment by subcatchment calibration is a very time-consuming effort, even if done in a semi-parallel way which is to our opinion not applicable for larger study regions such as on the national or continental level. A solution might be to switch to a land-characteristics-based universal method or to use a multi-scale parameter regionalization method as, for example, described in Mizukami et al. (2017).

For the fully-coupled WRF-Hydro run (WRF-H_FC), to obtain meaningful results, it was required to call the hydrological

part of the LSM at an hourly time-step, identical to that of the WRF-H_SA model. The problem can be attributed to numerical truncation effects in the overland routing routines when timesteps are in the order of a few seconds and spatial resolution is about 100 m. The most prominent impact of the enabled lateral routing, on (WRF-H_FC) versus (WRF_SA) is a general increase in soil moisture values due to lateral water transport and smaller REFKDT values and thus larger infiltration at the land surface which in turn leads to increased evapotranspiration for the summer months. Compared to the observations, the

coupled simulation performs better for most of the months and this finding holds also for the fluxes of sensible and ground heat. Solely for the mountainous site (DE-Ggw) both models show almost identical results which we attribute to generally higher soil moisture in that region and also to the reduced soil water storage capacity that comes with the decreased layer thicknesses defined for the slopy regions. In addition to the fluxes, also the near-surface states for air temperature and mixing ratio are better met with the fully-coupled model. The comparison with observed boundary layer temperature and humidity profiles at the DE-

Fen site gives also a higher rank for the WRF-H_FC simulation. WRF_SA yields much dryer soil moisture conditions, and in particular from July to September it cannot maintain the evapotranspiration that is seen from the observations. However, when compared with the SoilNet observations at the DE-Fen TERENO-preAlpine site, a considerable mismatch remains ascribable to the discrepancy of the soil maps used in the model and real world conditions. The STATSGO/FAO data assumes silty loam for almost the entire area of domain 3. This does not even rudimentarily reflect the complexity of the Alpine foothills, in

particular with the high resolutions of 1 km and 100 m of the atmospheric and hydrological model sections, respectively. The incorporation of recently available continental or global soil maps like, e.g., Hengl et al. (2017) and Tóth et al. (2017) could lead to further improvement here. The inter-model deviations for precipitation are marginal. The increased latent heat flux of the fully-coupled model does not strongly impact the precipitation generation for the study domain. It is rather assumed that the surplus in atmospheric moisture is actually being transported beyond the lateral boundaries. Thus, to see an impact of the

model coupling on precipitation patterns, the domain size should be sufficiently enlarged. It should be noted that, however, the impact of the lateral water transport modeling on average precipitation amounts may be much less important than, e.g., the selection of the PBL scheme (Arnault et al., 2018). The findings are also corroborated by water vapor tagging studies that conclude that regional precipitation recycling is largest during weak synoptic forcing and that variations on larger scales are rather small (Wei et al., 2015; Arnault et al., 2019). Nevertheless, for larger regions the impact may be considerable. Short-

and longwave radiation do not change much with the different model configurations. As for precipitation, the land–atmosphere





feedback of moisture by LSM–hydrological coupling has no noticeable impact on the cloud generation processes, thus leaving the biases unaffected.

The analysis of the subcatchment water budgets reveals a clear connection between the biases of precipitation, soil infiltration, evapotranspiration, and discharge. The other terms of the water balance equation, soil water storage variation and perco-
lation, do not show distinct trends between standard and coupled simulations which could be related to the inter-subcatchment variations of the infiltration and percolation parameters.

Using the calibrated parameters of WRF-H_SA in WRF-H_FC is required from a physical perspective. If deviation patterns reoccurred, a recalibration of WRF-H_FC could lead to improved discharge simulations, however at the risk of deteriorating the other water budgets.

In contrast to the uncoupled WRF model, fully-coupled simulations with an observation-calibrated hydrologically enhanced LSM show partly improved skill for land–atmosphere exchange variables, although the physical realism of the hydrological extension as well as for the spatial patterns of static data is still limited. Additional efforts are required to increase this physical realism, which should further improve the skill of the overall system. Including hydrological processes provides at least an additional way to calibrate and evaluate the simulations by taking also the regional water balance and budgets into account —
provided that comprehensive observations are available. Hopefully, it also enhances the joint modeling of water and energy exchange at the land–atmosphere boundary and helps to improve also future Earth System Modeling, like pointed out by Clark et al. (2015). To experience the full momentousness of coupled modeling future studies should consider continental extent. Also the sophistication in the description of the hydrological processes should be increased.

**Code and data availability**

The source code of the extended coupled WRF-3.7.1/WRF-Hydro-3.0 model, as used in this study, is available at Fersch (2019c, https://doi.org/10.5281/zenodo.3405780). The model configuration files (WRF and WRF-Hydro namelists) can be obtained from Fersch (2019a, https://doi.org/10.5281/zenodo.3407166). The TERENO-preApline, and the ScaleX campaign datasets used in this study are available for download at Fersch (2019b, https://doi.org/10.5281/zenodo.3406970). The discharge observations used for model calibration published by the Bavarian Environmental Agency at https://www.gkd.bayern.de. The
RADOLAN data of the German Weather Service (DWD) are available at https://opendata.dwd.de. The ASTER global digital elevation model can be obtained from https://doi.org/10.5067/ASTER/ASTGTM.003. ASTER GDEM is a product of Japan's Ministry of Economy, Trade, and Industry (METI) and NASA.

*Author contributions.* BF, AS, and HK developed the methodology for the study. AS conducted the initial WRF simulations and the PEST calibration of WRF-H_SA. BF implemented the changes to the WRF-Hydro model code and performed the WRF-H_SA and WRF-H_FC
simulations. BA, MM, KS, and IV conducted and processed the HATPRO, eddy-covariance, lysimeter, and soil moisture observations, respectively. BF and AS carried out the investigation and prepared the original draft with contributions from all authors. HK supervised the project.





*Competing interests.* The authors declare that they have no conflict of interest.

*Acknowledgements.* This work was funded by the Helmholtz Association Initiatives REKLIM, and TERENO, the Karlsruhe Institute of Technology, the University of Calabria, the German Academic Exchange Service (DAAD) and the German Research Foundation (DFG Research Unit *Cosmic Sense*). We are grateful to David Gochis and Wei Yu from National Center for Atmospheric Research, Boulder,
CO for their valuable suggestions and their assistance with the WRF-Hydro model. Moreover, we acknowledge the developers of the open source tools used for this study: GNU R (tidyverse, ggplot, sp, rgdal, influxdbr, stargazer, hydroGOF, hydroPSO, parallel), WRF, inkscape, TEX, Qgis, influxdb, PEST. The WRF-Hydro modeling system was developed at the National Center for Atmospheric Research (NCAR) through grants from the National Aeronautics and Space Administration (NASA) and the National Oceanic and Atmospheric Administration (NOAA). NCAR is sponsored by the United States National Science Foundation.





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
