# Peer review of "High-resolution fully-coupled atmospheric-hydrological modeling: a cross-compartment regional water and energy cycle evaluation"

_Hydrology and Earth System Sciences, 2019_

## Referee Comment (RC1) · Anonymous Referee #1 · 4 Nov 2019

Examining the ability of atmospheric-hydrological coupled modeling in the seasonal prediction is an important topic in hydroclimatic research. This study explored the ability of the WRF-hydro in capturing the regional hydrological cycle. Their results show considerable changes of latent and sensible fluxes between classic WRF and coupled WRF-Hydro though the calibrated parameters were same for both models. When compared to observations, the WRF-Hydro performed better than the classic WRF in the simulation for variety of variables including ET, sensible and ground heat flux, near surface mixing ratio and temperature. The value of this study lies on providing some information in the performance of WRF-Hydro that may be useful for weather/climate forecast service. I support it to be published in HESS if the following comments could

be addressed.

General comments:

1. For the Figure 2, since the parameter set was calibrated in WRF-Hydro (WRF-H_SA), we can expect that when coupled with atmospheric model, the WRF-Hydro (WRF-H_FC), which is the same model used in the calibration process, should have a better performance than the WRF standalone (WRF_SA) which is a different model to that used in the calibration process. Therefore, how could the authors demonstrate that the worse performance of WRF_SA is not caused by that the parameters calibrated in WRF-Hydro did not work well with WRF_SA?

2. While compared to WRF, the WRF-Hydro had a better performance, I am also curious about if compared to other classic hydrological models, such as SWAT, TOPmodel, how the performance would be for the WRF-Hydro, especially on the stream flow modeling. Though this may be out of the scope of the manuscript, adding such information (maybe by citing other previous research) could much add the value of this manuscript.

Specific comments:

P1, line 3: What are the traditional disciplines?

P8, line 13-14: Directly using soil moisture from WRF-H as an initial condition of WRF_SA can not guarantee the land surface has been well spun-up for the WRF_SA start.

P9, line 5: CML->CLM

P9, line 11-13: Usually, these processes should be included in a land surface model. Does the WRD-Hydro use more advanced (or complicated) representations for those processes replacing the old ones in the Noah-MP, or there are no such processes in the Noah-MP at all?

P10, line 18: Are there any reservoirs distributed in the basins? If so, why did you turn

off the reservoir module? What uncertainties may come from this setting?

P10, line 24-25: Please explain the meaning of "two and three dimensions of soil features representation".

P12, line 12-13: What is this setting of REFKDT used for?

P13, line 1-2: Does this simplification induce some errors in regarding to the land-atmospheric interactions?

P14, figure 4: Why there are no shifted hydrographs for Ach-OBH, Ach-OBN and Rt-RST?

P14, line 9: Could the authors show the net radiation for both observations and simulations?

P15, figure 5: Same as P14, figure 4.

P29, line 12-13: How does the lateral water transport increase the soil moisture?

P29, line 16-18: It seems to me that the decrease of soil thicknesses made the results worse in the mountainous site. So why did you change the soil layers for the slope region?

P30, line 16: In terms of Earth System Modeling, how to calibrate the Earth System Model globally is quite challenging. Observation data are so limited on this scale.

---

## Referee Comment (RC2) · Anonymous Referee #2 · 25 Nov 2019

Review of " Fersch et al. - High-resolution fully-coupled atmospheric–hydrological modeling: a cross-compartment regional water and energy cycle evaluation" The authors present a comprehensive effort in setting up, running and validating a WRF-hydro model run and comparing it to a non-hydro WRF setup. I have suggested major revisions and would like to see the following addressed before publication, but I would like to highlight, that with the changes made, the manuscript is indeed worthy of publication. I hope the comments can also be seen as improvements in readability.

Introduction: P2L2-4: The introduction seems to have an abrupt start. Does this first paragraph really have the phrasing to initially frame the story? At least, add ". . . and

magnitude..." after "spatial distribution".

P3L1: The newest general HOBE publication is "Jensen, K.H., Refsgaard, J.C. (2018): HOBE – the Danish Hydrological Observatory, Vadose Zone Journal, 17:180059, doi:10.2136/vzj2018.03.0059" This could be changed or added.

General intro: I believe an entire paragraph is missing here where the following is addressed: - What is the key relevance of this paper to the public and/or research domain? Why are you/we doing this? This is important but often overlooked. - What is new compared to previous Fersch/South-German studies as well as other studies? A lot of work is mentioned, but the relevance and new story here is overlooked. Please relate.

P7L8: Radolan reference?

P7L9-10: What is the uncertainty in doing this compared to using observations?

P7L3 to P8L6: I think this sections needs a paragraph that explains the key differences between WRF classic, WRF-H-SA and WRF-H-FC to be able to better understand the reasoning here.

P8L5: This "parameter set" has not yet been explained and confuses. What are these parameters? They belong to the atmospheric realm I believe and are therefore not hydrologic calibration parameters.

P9L2-11: Really hard to link between text and table (1). The schemes/physics as well as references do not match up and several domain levels are refereed to. Please elaborate/correct.

P9L17: "only subsurface and surface overland flow routing is..." -> ok, but then state what is not taken into account in the atmosphere link – the non-WRF-hydro expert does not know this (me included), but would have to guess.

Figure 1: Many site abbreviations, which are not really 'learned' when reading the

manuscript. I don't know if a systemativ naming approach could be thought of.

P10L26-27: "Furthermore. . ." -> How was this done? Based on what? Which data?

P11L28: You could also add: Larsen, M.A.D, Refsgaard. J.C., Jensen, K.H., Butts, M.B., Stisen, S., Mollerup, M. (2016): Calibration of a distributed hydrology and land surface model using energy flux measurements, Agricultural and Forest Meteorology, 217, 74-88, doi:10.1016/j.agrformet.2015.11.012. And; Stisen, S., McCabe, M.F., Refsgaard, J.C., Lerer, S., Butts, M.B. (2011): Model parameter analysis using remotely sensed pattern information in a multi-constraint framework. Journal of Hydrology, 409, 337-349. doi: 10.1016/j.jhydrol.2011.08.030.

P12L17: Computational resources -> should these not be mentioned in the manuscript?

P12L20-P13L1: Why/how did you add the baseflow. I see it was necessary, but please elaborate why/how etc. Also – please link the the "_sh" runs in fig 4 + 5.

Table 3: For Am-PEI "coeff" and "expon" the values are larger than the suggest span in table 2.

Table 3: For the two parameters mentioned above as well as zmax = 1 (3 instances) and retdp ) = 5, the autocalibration seems to have hit a boundary limit, which could imply model deficiencies. Please mention/elaborate or re-run, using more sound limits.

P13L10: First sentence: Not understood. Limited – how?

Table 4-9: I think you should add MAE.

P20L6-13: Isn't these just the latent heat results again? They look the same. If not, how are they different and why?

P20L6-9: Unclear which scenario is most realistic? Why not just use the best? Better word than scenario?

P22L3-4: Please see my previous comment on table 3 about the resulting parameters. "...often tuned to unrealistic values..."

P22L14-15: Why? Dicuss/relate! This is a discussions section.

P26L19: Please write "(NSE)" after Nash Sutcliffe efficiency.

P28L8-9: Why was this the case?

P29L2: "For the validation period" -> has this been addressed previously?

P29L9-10: Why is this a problem (the 1h time steps)? If it corresponds to "SA" mode.

P29L24: Reference? On alpine foothills soil texture.

P30L15: I do not like the use of "hopefully". If you had framed the relevance of the study in the introduction, then this paragraph would be more easily written.

Minor: Abstract: "Nominal" – use better wording?

P12L1: where "the" model. Sounds better to my taste.

P26L25-26: "difficulty in reproducing" instead.

---

## Referee Comment (RC3) · Anonymous Referee #3 · 26 Nov 2019

With the development of fully-coupled atmospheric-hydrological model especially the improvement in the hydrological part, whether the improvement in land surface process (e.g., hydrological process) can lead to a better simulation of regional water and energy cycle is an issue that is concerned by the whole literature. Fersch et al., presents a very comprehensive research on this issue by comparing the coupled model (WRF-Hydro) with the original one (WRF_SA). Generally, their experiment is well designed, the results are convincing and the research does have added value to deepen our understanding on the performance of fully-coupled atmospheric–hydrological modeling. I think it can be published in HESS after addressing the following comments: Major comments: 1.Introduction of WRF-Hydro. From the introduction of the WRF-Hydro, it

seems that WRF-Hydro improves the physical description of surface flow and baseflow transportation. But do the WRF-Hydro and WRF_SA have the same runoff scheme? Will and how the surface/subsurface lateral flow feed back to the land (e.g., through re-infiltration)? I suggest the author give some introduction on the difference of basic WRF and WRF-Hydro as not all readers are familiar with this model. 2.Meaning of different parameters. The current interpretation of physical meaning of parameters is distributed. For example, in P12,L12 "The LSM surface runoff scaling parameter REFKDT is globally set to $2e-06$ as smaller values would have decreased infiltration to very small amounts." P28,L12"Also the percolation parameter SLOPE was mainly reduced as compared to the standard value, meaning that a considerable portion of former infiltration excess water needed to be transferred to the bucket-storage to assure good performance for the simulated baseflow". They should be directly introduced in the calibration section. Moreover, I suggest a brief interpretation of the physical meaning of changes of parameter. 3.What is the uncertainty of observations especially for the evapotranspiration? Does the improvement exceed the uncertainty?

Major comments: 1.P10, L25: What do the "two dimensional" and "three dimensional" mean? Does the "soil layer depths" mean the "soil layer thickness"? 2.P12, L12: The LSM sets the REFKDT as 0.2e-6, this value is not within the range given in the Table 2. 3.section 2.4.6: Please give the timescale of calibration and validation. P13,L4-8: If you do not include the abnormally high value, which NSE efficiency you can get? Will it comparable to that in calibration period. 4.P26,L8: What do the 50% and 100% mean? Which reference they refer to? 5.P26,L17: Just a suggestion. Will the results improve if you use daily streamflow, as the hourly streamflow is difficult to simulate? 6.P29,L12: Again, I do not know how does the lateral water transport saturate the soil as you do not introduce this in the model introduction section.

---

## Author Response (AR1)

**Response to the reviewers for hess-2019-478**

High-resolution fully-coupled atmospheric–hydrological modeling: a cross-compartment regional water and energy cycle evaluation

Benjamin Fersch, Alfonso Senatore, Bianca Adler, Joël Arnault, Matthias Mauder, Katrin Schneider, Ingo Völksch, and Harald Kunstmann

The authors would like to thank the referees for their interest in the manuscript and their willingness to provide their valuable comments. A point by point discussion of the issues is enclosed below. Since several identical points were made by different referees, we decided to compile all answers into one document and to hyper-link among the related comments in the PDF file.
* * *
**REFEREE # 1**

Examining the ability of atmospheric-hydrological coupled modeling in the seasonal prediction is an important topic in hydroclimatic research. This study explored the ability of the WRF-hydro in capturing the regional hydrological cycle. Their results show considerable changes of latent and sensible fluxes between classic WRF and coupled WRF-Hydro though the calibrated parameters were same for both models. When compared to observations, the WRF-Hydro performed better than the classic WRF in the simulation for variety of variables including ET, sensible and ground heat flux, near surface mixing ratio and temperature. The value of this study lies on providing some information in the performance of WRF-Hydro that may be useful for weather/climate forecast service. I support it to be published in HESS if the following comments could be addressed.
* * *
**COMMENT # 1.1**

For the Figure 2, since the parameter set was calibrated in WRF-Hydro (WRF-H_SA), we can expect that when coupled with atmospheric model, the WRF-Hydro (WRF-H_FC), which is the same model used in the calibration process, should have a better performance than the WRF standalone (WRF_SA) which is a different model to that

used in the calibration process. Therefore, how could the authors demonstrate that the worse performance of WRF_SA is not caused by that the parameters calibrated in WRF-Hydro did not work well with WRF_SA?

**Answer:**

We would like to thank the referee for this comment, which offers the opportunity to clarify two important aspects related to fully-coupled modeling compared to uncoupled, which goes beyond the 'mere' performance comparison in a case study.

The first aspect is related to the fact that, in our experiments, WRF_SA and WRF-H_FC share exactly the same parameters: in agreement with previous literature, the results highlight that, with the same parameters, the fully-coupled model provides wetter soil conditions than the uncoupled. In our opinion, from a point of view of hydrological processes understanding, this result is quite interesting, because highlights that, discarding the (real) soil water redistribution processes with uncoupled model is a possible cause obliging to tune soil parameters to "unrealistic values for the sake of obtaining a good matching", as we wrote.

The second aspect concerns a methodological issue. The calibrated parameter set is achieved using a very peculiar variable, i.e. the discharge at the outlet, which summarizes the hydrological behavior of the whole upstream area. In other words, calibrating some LSM parameters using the hydrological model allows to take into account representative features of a wide (catchment) area, summarized in the discharge variable. On the other hand, the parameterization of the same parameters using only the WRF_SA model would be less straightforward. Of course, they could be parameterized with respect to point observations (e.g., latent/heat fluxes or soil moisture measurements), which however do not guarantee the same accuracy in the remaining area. Also, a two-dimensional spatially distributed calibration could be attempted, e.g., with precipitation (whose spatial distribution is however always provided by other models, e.g., spatial interpolations, reflectivity-precipitation conversion formulas, etc.), which however would result much more difficult than the calibration based on the hydrological model. If this point is considered, the better results achieved with WRF-H_FC in our experiments with flux and soil moisture observations are even more valuable. Nevertheless, it is clearly possible that other parameter combinations for WRF_SA could lead to likewise or even better performance as with WRF-H_FC.

We have changed the following parts of the manuscript to incorporate this discussion:

Update in the **Introduction**:

Our study presents a concept to improve the physical realism of regional dynamical hydrometeorological simulations not only by taking into account lateral water redistribution processes on the land surface and their coupled feedback with the planetary boundary layer but also by evaluating the simulated water and energy budgets with comprehensive observations. We calibrate the land surface model that is used in the coupled modeling system based on discharge observations of several subcatchments and thus rely on a variable that integrates the hydrological behavior of the whole upstream area. In a classic local area modeling study, we could only tune land surface parameters based on station observations which would be less straightforward with respect to the different scales of simulation and observation. We investigate how well the hydrologically enhanced, fully-coupled model mimics observations for different compartments of the hydrological and the associated energy cycle.

Update in the **Modeling chain** subsection:

Consequently, WRF_SA does not represent an optimized setup of a classic WRF stand alone model. Therefore, it is possible that with other parameter combinations for WRF_SA even better performance could be achieved. However, tuning WRF_SA is not easily possible because it does not feature the simulation of discharge and other point observations are sparse and represent different scales and are thus only suitable for the evaluation of simulation results.

Update in the **Summary and conclusive remarks** section:

The evaluation included the standalone WRF (WRF_SA) and the fully-coupled (WRF-H_FC) models that share identical parameter sets, initial and boundary conditions for a commensurable set of simulations. It is of course possible that other parameter combinations for WRF_SA could lead to likewise or even better performance as with WRF-H_FC. E.g., the dryer soils could be alleviated by increasing the value of the infiltration parameter REFKDT which in turn would increase evapotranspiration and decrease sensible heat. WRF_SAcould be parameterized with domain wide setting with respect to point observations (e.g., latent/heat fluxes or soil moisture measurements), which however do not guarantee the same accuracy in the remaining area and the effects of the intermediate surface water storage and lateral flow that are particularly important for strong precipitation events would not be considered. In contrast, the hydrological model allows to take into account representative features of a wide (catchment) area, summarized in the discharge variable.

COMMENT # 1.2

While compared to WRF, the WRF-Hydro had a better performance, I am also curious about if compared to other classic hydrological models, such as SWAT, TOPmodel, how the performance would be for the WRF-Hydro, especially on the stream flow modeling. Though this may be out of the scope of the manuscript, adding such information (maybe by citing other previous research) could much add the value of this manuscript.

We have added available information to relate our results to other studies with WRF-Hydro in general and other hydrological models for the Ammer catchment. In the model description section, we wrote:

The results of the calibration are in line with other hydrological modeling studies for the Ammer catchment. Ludwig and Mauser (2000) implemented TOPMODEL (Beven et al., 1984) into a SVAT model framework and yielded a NSE of 0.92 for one year simulation on a daily basis for the gauge Fischen (nearby AM-WM). Marx (2007) achieved NSE performances of 0.2 (Am-OAG), 0.42 (Am-PEI), 0.75 (Am-WM), 0.68 (Ach-OBN), and 0.18 (Ach-OBH), using WaSiM-ETH (Schulla and Jasper, 2007) for the year 2001. Rummler et al. (2018) obtained a NSE of 0.91 with the WRF-Hydro standalone model for Am-PEI, for a 3 months simulation of a major flood event in 2005.
* * *
COMMENT # 1.3

P1, line 3: What are the traditional disciplines?

**Answer:**
With the wording "traditional disciplines ", we wanted to express that studies are often restricted to the viewpoints of certain scientific disciplines such as meteorology or hydrology. We have changed the sentence to

Modeling instead is often narrowed to single compartments of the terrestrial system or bound to traditional viewpoints of definite scientific disciplines.
* * *
COMMENT # 1.4

P8, line 13-14: Directly using soil moisture from WRF-H as an initial condition of WRF_SA can not guarantee the land surface has been well spun-up for the WRF_SA start.

**Answer:**
The spin-up of the land surface (canopy, and first soil layers) is usually a matter of hours to days. This is taken into account by having another 15 days of spin-up (April 15 to May 01 2016). For the deeper soil, we agree that the system may not be in full equilibrium, however, we assume that the conditions in soil layers 3 and 4 are more realistic with the 6 month + 15 days spin-up than with a 15 days only spin-up. It is important that both models WRF-H_SA and WRF-H_FC start with identical conditions, so that the differences between standalone and coupled mode are commensurable. The motivation for this spin-up strategy is to avoid the uncertain simulaton of the snow storage dynamics over the winter season.

We updated the space and time subsection as follows:

For the surface variables in WRF-Hydro we consider a 15 day spin-up to be sufficient. The initial (2016-04-15) soil moisture fields for both WRF_SA and WRF-H_FC are taken from the last WRF-H_SA simulation timestep (2016-10-31). We assume that this 6 month spin-up period is sufficient to come up with reasonable starting conditions for a commensurable set of simulations.
* * *
COMMENT # 1.5

P9, line 5: CML→CLM

**Answer:**
Changed.
* * *
COMMENT # 1.6

P9, line 11-13: Usually, these processes should be included in a land surface model. Does the WRF-Hydro use more advanced (or complicated) representations for those processes replacing the old ones in the Noah-MP, or there are no such processes in the Noah-MP at all?

**Answer:**

In Noah MP it is possible to select a TOPMODEL based surface runoff parameterization and furthermore, a column based confined groundwater model can be activated. Whereas the groundwater model enlarges the vadoze zone storage body of the classic Noah model to the phreatic zone and thus provides an overall increased storage body, the TOPMODEL scheme does not offer an option for horizontal redistribution of infiltration excess water which is analogous to the Schake et al. model in Noah-MP that was used in this study. The surface and groundwater runoff produced by Noah MP can be used to drive separate runoff and channel routing tools. What WRF-Hydro uniquely provides is the overland routing of surface runoff that allows for reinfiltration and thus lateral redistribution of water on the land surface thus enabling feedback with the boundary layer. We slightly changed the paragraph in the manuscript to

Noah-MP provides different options for the computation of groundwater discharge and surface runoff (infiltration excess) but only WRF-Hydro enables the simulation of lateral hydrological processes such as the overland routing of surface runoff and channel (discharge) routing.

COMMENT # 1.7

P10, line 18: Are there any reservoirs distributed in the basins? If so, why did you turn off the reservoir module? What uncertainties may come from this setting?

**Answer:**

There is one single lake contained in the Ach subcatchment (Staffelsee, 7.66 km$^2$). The gauge Ach-OBN is located upstream and Ach-OBH is located downstream of the lake. From the observations we see that the lake dampens the hydrograph a bit with leads to reduced peaks and a slightly increased baseflow. When we look at the calibration and validation hydrographs, we see that upstream and downstream gauges are simulated reasonably. Therefore, we refrained from adding the reservoir module to the simulation as this would have further increased the number of parameters and thus the amount of calibration runs whereas the effect on the hydrographs are anticipated to be only minor. Since the lake evaporation is separately considered by the land surface model, we do not assume any impact on the land surface–boundary layer exchange. We updated the statement to expound our motivation as follows:

The reservoir module (retention of channel flow in lakes and reservoirs) is disabled for the simulations of this study because only one gauge (Ach-OBH) would be slightly affected by a 7.66 km$^2$ lake (Staffelsee) but the number of calibration parameters and thus calibration runs would considerably increase. Since the lake evaporation is explicitly considered by the Noah-MP land surface model we do not assume any impact on the land surface–boundary layer exchange by this simplification.
* * *
COMMENT # 1.8

P10, line 24-25: Please explain the meaning of "two and three dimensions of soil features representation".

**Answer:**
In its standard version, Noah-MP supports only the domain wide assignment of soil layer thicknesses (typically set to 10, 30, 60, and 100 cm, for layers one to four), with 200 cm total depth. We have changed this domain-wide setting to a three-dimensional array to support variable soil thickness at each grid point (please refer to Comment 1.15 for the discussion about the benefit of variable depth soil layers). The new implementation provides also the flexibility to use three dimensional information for other soil features, such as e.g., maximum soil moisture content or saturated hydraulic conductivity. However, we did not make use of this feature because it would have required three dimensional soil maps but these were not available. To make this more clear in the manuscript, we changed the text to:

Here, the model's general assumption of two meter soil thickness (or depth to bedrock) does not hold true, as it may lead to overestimated retention of infiltrating water. Therefore, the soil layer thickness definition was changed from a domain uniform to a grid point based representation and soil layer depths were set to the Noah-MP standard (0.1, 0.3, 0.6, 1 m) distribution for hillslopes below 50 % and to more shallow values (0.05, 0.05, 0.1, 0.1 m), for all slopes >50 %. The 50 % threshold led to a realistic discriminability of valley bottoms and hillslopes.
* * *
COMMENT # 1.9

P12, line 12-13: What is this setting of REFKDT used for?

Related with Comment 3.2 and Comment 3.5.

**Answer:**
The REFKDT parameter is erroneously mentioned here. Instead the parameter addressed in this sentence is REFDK which is the scaling parameter for the infiltration equation. We have set this in the simulations to the Noah-MP documented standard value which is $2e^{-06}$ and the typo in the text was corrected.
* * *
COMMENT # 1.10

P13, line 1-2: Does this simplification induce some errors in regarding to the land-atmospheric interactions?

Related with Comment 1.11 and Comment 2.14.
**Answer:**
The underestimation of baseflow is not relevant for the land surface–boundary layer exchange, since channel water does not interact with the LSM (channel loss is not considered). We added one sentence to the respective section in the manuscript:

> The underestimation of long-term baseflow by the model has no influence on the land surface–atmosphere exchange as there is no interaction of the channel routing with the LSM.
* * *
COMMENT # 1.11

P14, figure 4: Why there are no shifted hydrographs for Ach-OBH, Ach-OBN and Rt-RST?

Related with Comment 2.14
**Answer:**
Baseflow shifting is motivated by the hydrogeological conditions of the respective subcatchments. The detailed answer to this comment is provided with the answer for Comment 2.14.
* * *
COMMENT # 1.12

P14, line 9: Could the authors show the net radiation for both observations and simulations?

**Answer:**
While the variables lw and sw allow to get a direct comparison, a net radiation comparison is less straightforward, since it could lead to "right results for the wrong reason" (e.g., compensation of under- and over-estimations in the variables of the radiation balance).
* * *
COMMENT # 1.13

P15, figure 5: Same as P14, figure 4.

**Answer:**
Figure 4 depicts the calibration period whereas Figure 5 shows the validation period. We added the calibration period dates to the caption of Figure 4 to make this more clear.
* * *
COMMENT # 1.14

P29, line 12-13: How does the lateral water transport increase the soil moisture?

Related with Comment 3.9.
**Answer:**
The infiltration excess computed by the LSM is transferred to a surface storage layer in WRF–Hydro. Thus, the surface runoff remains in place which allows for subsequent infiltration or becomes horizontally routed by the overland routing module once the retention depth of the surface is exceeded. Thus, the water remains on the surface until it infiltrates, evaporates or reaches a channel element. We have added some more detail in the model description section:

WRF-Hydro (Gochis et al., 2016) augments WRF with respect to lateral hydrological processes at and below the land surface. It adds a surface storage layer where infiltration excess water is stored and subsequently routed according to the topographic gradient once the retention depth becomes exceeded. This is different to WRF where the infiltration excess depicts a sink term. Thus, in WRF-Hydro the surface water can infiltrate gradually, potentially leading to increased soil moisture. Gradient based routing can also be activated for saturated soil layers, and in case of oversaturation

the water will exfiltrate to the surface where it enters the surface storage body and routing process. WRF-Hydro is connected with the planetary boundary layer in the same way as WRF, the lateral water transport at and below the surface is the crucial difference. Further hydrological processes that are implemented in WRF-Hydro without feedback to the atmosphere are baseflow generation and channel routing. The model has two operation modes, stand alone, driven by gridded (pesudo-) observations (one-way coupled) or fully coupled with the dynamic atmopheric model WRF (fully-coupled).
* * *
COMMENT # 1.15

P29, line 16-18: It seems to me that the decrease of soil thicknesses made the results worse in the mountainous site. So why did you change the soil layers for the slope region?

Related with Comment 2.25 and Comment 3.4.

**Answer:**

When we had interpreted the simulated WRF-H_SA hydrographs for the mountainous site (Am-OAG) we found that most of the observed peaks were missing whereas post-peak flow exceeded the measurements. We concluded that this is caused by too much retention of the effective precipitation on the mountain slopes. The 2 m soils on 50% sloped soils is not realistic, typically the bedrock layer starts between 10 and 50 cm depth (see also Ludwig et al., 2003; Hofmann et al., 2009, for a description of soils in the Ammer catchment region). Naturally, the assumption of a hard $22.5°$ threshold is not quite realistic. However, it gave the strongest gain in performance for our region among the range of values that we tested (between 20 and 60% slope). One could also think of a gradient dependent function that blends from the 2-m standard profile to the thin slope profile. However, this would be a study of its own. Figure 1 and 2 show the differences for standard and alternative soil profile distribution. Most affected are hydrographs for Am-OAG and Am-PEI, they also hold the largest proportion of steep slopes. Thus in terms of water balance, this quick runoff from the mountain slopes is an important process that needs to be considered in the model. We have added some lines to the calibration section and two figure to the supplementary material to explain this a bit more in detail:

The optimum slope gradient for the delineation of shallow and deep soil regions is found to be 50% ($22.5°$). Accounting for the shallow mountain slopes considerably improves the hydrographs for Am-OAG and Am-PEI, as it increases underestimated peaks and decreases overestimated retention. A comparison of the respective hydro-

[Figure]

Figure 1: Simulated and observed discharge for calibration period (2015-05-01–2015-07-31. (a) Uniform soil layers throughout domain, (b) reduced soil layer thicknesses for slopes > 50%.

[Figure]

Figure 2: Simulated and observed discharge for calibration period (2016-05-01–2016-10-31. (a) Uniform soil layers throughout domain, (b) reduced soil layer thicknesses for slopes > 50%.

graphs and performance measures for calibration and validation period is shown in figures S1 and S2.
* * *
COMMENT # 1.16

P30, line 16: In terms of Earth System Modeling, how to calibrate the Earth System Model globally is quite challenging. Observation data are so limited on this scale.

Related with Comment 2.26.
**Answer:**
We agree with the referee that limited data and also computational constraints are a challenge for setting up a model like WRF-Hydro for global application. With our statement, we refer more to regional applications or up to continental extent which seems feasible. We updated the paragraph also with respect to the comment of reviewer #2.

The combined approach offers potential to improve also future Earth System Modeling, like also pointed out by Clark et al. (2015). To experience the full momentousness of coupled atmospheric– modeling future studies should be extended to larger regions to cover the scales of atmospheric recycling processes. Also the descriptions of the hydrological processes in the models should be further refined as computational capabilities increase and with more and more detailed data products becoming available.
* * *
Review of "Fersch et al. - High-resolution fully-coupled atmospheric–hydrological modeling: a cross-compartment regional water and energy cycle evaluation". The authors present a comprehensive effort in setting up, running and validating a WRF-hydro model run and comparing it to a non-hydro WRF setup. I have suggested major revisions and would like to see the following addressed before publication, but I would like to highlight, that with the changes made, the manuscript is indeed worthy of publication. I hope the comments can also be seen as improvements in readability.

COMMENT # 2.1

Introduction: P2L2-4: The introduction seems to have an abrupt start. Does this first paragraph really have the phrasing to initially frame the story? At least, add "... and magnitude..."after "spatial distribution".

**Answer:**
We updated the introduction for a more intuitive start for the story. We added this paragraph at the beginning:

The intertwined exchange of water and energy fluxes at the land–atmosphere interface determines hydrological processes on a multitude of spatial and temporal scales. Its appropriate formulation and implementation into model systems is a prerequisite for climate- and land use change impact investigations. Both, terrestrial and atmospheric processes need to be considered. Fully coupled hydrological–atmospheric model system have recently been developed and comprise the most relevant Earth system components. Comprehensive and concerted evaluation of these coupled modeling systems is required to assess the current limits and potential in Earth system science. This study accordingly focuses on the evaluation of a fully-coupled atmospheric–hydrological model across the various compartments of the water and energy cycle.

COMMENT # 2.2

P3L1: The newest general HOBE publication is "Jensen, K.H., Refsgaard, J.C. (2018):

HOBE – the Danish Hydrological Observatory, Vadose Zone Journal, 17:180059, doi:10.2136/vzj2018.03.0059". This could be changed or added.

**Answer:**
We have replaced the HOBE reference with the latest one.

The most prominent activities for Europe are HOAL (Hydrological Open Air Observatory, Blöschl et al., 2016), HOBE (The Danish Hydrological Observatory, Jensen and Refsgaard, 2018), LAFO (Land-atmosphere feedback observatory, Spath et al., 2018), and TERENO (TERestrial ENvironmental Observatories, Zacharias et al., 2011).
* * *
COMMENT # 2.3

General intro: I believe an entire paragraph is missing here where the following is addressed: - What is the key relevance of this paper to the public and/or research domain? Why are you/we doing this? This is important but often overlooked. - What is new compared to previous Fersch/South-German studies as well as other studies? A lot of work is mentioned, but the relevance and new story here is overlooked. Please relate.

**Answer:**
We have updated the introduction section with the following paragraph to express the novelty of the approach

Our study presents a concept to improve the physical realism of regional dynamical hydrometeorological simulations not only by taking into account lateral water redistribution processes on the land surface and their coupled feedback with the planetary boundary layer but also by evaluating the simulated water and energy budgets with comprehensive observations. We calibrate the land surface model that is used in the coupled modeling system based on discharge observations of several subcatchments and thus rely on a variable that integrates the hydrological behavior of the whole upstream area. In a classic local area modeling study, we could only tune land surface parameters based on station observations which would be less straightforward with respect to the different scales of simulation and observation. We investigate how well the hydrologically enhanced, fully-coupled model mimics observations for different compartments of the hydrological and the associated energy cycle.
* * *
COMMENT # 2.4

P7L8: Radolan reference?

**Answer:**
We have added the reference for RADOLAN to the modeling chain section, at the first occurrence of the acronym in the text.

Altogether, as outlined in Figure 2, the modeling chain encompasses the following four steps: 1) a classic standalone WRF run (2015-04-01–2016-10-31) with standard LSM parameters to derive driving variables for 2) the standalone WRF-Hydro simulations (WRF-H_SA, 2015-04-01–2015-07-31 and 2016-04-15–2016-10-31) that ingest also gridded observed precipitation (RADOLAN, Bartels et al., 2004; Winterrath et al., 2012), 3) a fully coupled WRF-Hydro simulation (WRF-H_FC 2016-04-15–2016-10-31) using calibrated parameters from WRF-H_SA, and 4) a rerun of the classic standalone WRF (WRF_SA, 2016-04-15–2016-10-31) with the same parameter set obtained from WRF-H_SA.

COMMENT # 2.5

P7L9-10: What is the uncertainty in doing this compared to using observations?

**Answer:**
For the Ammer catchment, or the region in general, consistent station observations of all required variables are not available. For selected variables few stations are spread throughout the domain but because of the complex terrain, geostatistical interpolation would very likely lead to unrealistic spatial patterns of the driving variables. Moreover, only for precipitation, gridded data products are available with the required resolution. As the focus with WRF-H_SA is on discharge calibration we assume that the high resolution gridded RADOLAN precipitation product is most important to obtain good results. Deviations in any of the other driving variables would probably lead to variations in evapotranspiration and could probably affect the recession and baseflow components of the hydrographs.

COMMENT # 2.6

P7L3 to P8L6: I think this sections needs a paragraph that explains the key differences between WRF classic, WRF-H-SA and WRF-H-FC to be able to better understand the

reasoning here.

Related with Comment 3.1.
**Answer:**
We added a paragraph to the section *Model setup and calibration* to make the differences between the different model setups clear:

> To finally come up with a fully coupled WRF-Hydro setup several intermediate steps are required that involve different components of the modeling system. As outlined in Fig. 2 we build a modeling chain with the items WRF-Hydro Standalone (WRF-H_SA), WRF Standalone (WRF_SA), and WRF-Hydro Fully-Coupled (WRF-H_FC). WRF-H_SA refers to the hydrologically extended land-surface model that is not coupled to an atmospheric model and gets its driving data from gridded (pseudo-) observations. WRF Standalone (WRF_SA) is the classic version of WRF that has no hydrological extension and that is driven by data from a global circulation model. WRF-H_FC extends WRF_SA with the hydrological implementations of WRF-H_SA.
* * *
COMMENT # 2.7

> P8L5: This "parameter set" has not yet been explained and confuses. What are these parameters? They belong to the atmospheric realm I believe and are therefore not hydrologic calibration parameters.

**Answer:**
The new run of WRF-SA with the calibrated parameter set is needed to achieve our commensurable set of simulations for further comparison, since some of the calibrated parameters of the standalone hydrological model affect also the LSM (which is integrated in the WRF modelling system). We agree with the reviewer that this paragraph can be misleading if the parameter set is not known yet. However, for the sake of clarity and simplicity, we preferred to go into details of it later, when describing the Calibration procedure, rather than here, where the general structure of the experiment is presented. Therefore, we changed the text here to:

> The modeling chain encompasses the following four steps: 1) a classic standalone WRF run (2015-04-01–2016-10-31) with standard LSM parameters to derive driving variables for 2) the standalone WRF-Hydro simulations (WRF-H_SA, 2015-04-01–2015-07-31 and 2016-04-15–2016-10-31) that ingest also gridded observed precipitation (RADOLAN, Bartels et al., 2004; Winterrath et al., 2012). At this stage, a calibrated set of parameters influencing the overland flow routing, the baseflow model and the LSM is achieved; 3) a fully coupled WRF-Hydro simulation (WRF-H_FC 2016-04-15–2016-10-31) using

calibrated parameters from WRF-H_SA, and 4) a rerun of the classic standalone WRF (WRF_SA, 2016-04-15–2016-10-31) with the same calibrated LSM parameters obtained from WRF-H_SA. Finally, this leads to a commensurable set of simulations, coupled versus uncoupled.
* * *
COMMENT # 2.8

P9L2-11: Really hard to link between text and table (1). The schemes/physics as well as references do not match up and several domain levels are refereed to. Please elaborate/correct.

**Answer:**
In table 1, we list the configuration of the physical schemes for the WRF model in the WRF_SA and WRF-H_FC simulation runs. We do not want to repeat all schemes in the text again. Only for cumulus and land surface model we added some important additional information. We have updated the paragraphs to give it a better structuring:

The WRF physics parameterization for the selected domains is listed in Tab. 1. Cumulus parameterization is only activated for the outermost domain (1), while explicit convection is chosen for the finer grids (domain 2 & 3), according to Skamarock et al. (2008).

For uncoupled and coupled simulations, Noah-MP (Niu et al., 2011) is used as the land surface model. For Noah-MP the selected configuration deviates from the default setup as follows: the Community Land Model (CLM) method is selected for stomatal resistance computation, the Schaake et al. (1996) method is used to determine infiltration and drainage (similar to classic Noah-LSM), the two stream radiation transfer applied to vegetated fraction (option 3) is chosen, and the dynamic vegetation option (Dickinson et al., 1998) is enabled.

Furthermore, we have updated the caption of Table 1 as follows:

**Table1.** Selected WRF_SA and WRF-H_FC model physical schemes.
* * *
COMMENT # 2.9

P9L17: "only subsurface and surface overland flow routing is..." → ok, but then state what is not taken into account in the atmosphere link – the non-WRF-hydro expert

does not know this (me included), but would have to guess.

**Answer:**
The second paragraph about WRF-Hydro (starting with *Concerning the one-way...*)
describes the processes that have no feedback with the atmospheric part of the fully
coupled model. We changed the first sentence of the paragraphs so that it clearly
refers as antagonist with the previous one:

Concerning the one-way coupled processes modeled in the WRF-Hydro system (i.e.,
no feedback with the atmosphere), in this study, the channel flow and baseflow mod-
ules are used.

COMMENT # 2.10

Figure 1: Many site abbreviations, which are not really 'learned' when reading the
manuscript. I don't know if a systematic naming approach could be thought of.

**Answer:**
The naming of the stations and locations follows the following scheme: for the TERENO
Pre-Alpine sites they are the official site codes (also used in other flux databeses,
e.g., ICOS network). The discharge gauges consist of an abbreviation for the channel
(Rt=Rott, Ach=Ach, Am=Ammer) and the location of the gauges. So to our opinion,
the naming scheme for the stations is already systematic and we would like to keep
it this way. To make the figure slightly clearer, we have removed the ScaleX-tag from
Figure 1b.

COMMENT # 2.11

P10L26-27: "Furthermore..."– How was this done? Based on what? Which data?

**Answer:**
In the original versions of WRF and Noah-MP the infiltration and percolation pa-
rameters (REFKDT and SLOPE) are read from a text file and are valid for the whole
domain. In our version, we read the parameters from a 2-d netCDF fields, however,
the values are uniform for each of the sub-catchments (lumped). We have rephrased
the sentence in the manuscript to make this more clear:

In addition, the infiltration (REFKDT) and percolation (SLOPE) parameter implementation was changed from domain-wide uniform values to subcatchment-wise distributed (lumped).
* * *
COMMENT # 2.12

P11L28: You could also add: Larsen, M.A.D, Refsgaard. J.C., Jensen, K.H., Butts, M.B., Stisen, S., Mollerup, M. (2016): Calibration of a distributed hydrology and land surface model using energy flux measurements, Agricultural and Forest Meteorology, 217, 74-88, doi:10.1016/j.agrformet.2015.11.012. And; Stisen, S., McCabe, M.F., Refsgaard, J.C., Lerer, S., Butts, M.B. (2011): Model parameter analysis using remotely sensed pattern information in a multi-constraint framework. Journal of Hydrology, 409, 337-349. doi: 10.1016/j.jhydrol.2011.08.030.

**Answer:**
The two papers were added to the list.

Several approaches are adopted to reduce or control this problem, particularly challenging for the emerging fully distributed paradigm in hydrology (Beven and Binley, 1992; Beven, 2001; Kelleher et al., 2017), either constraining the parameter set by means of various strategies (e.g., Cervarolo et al., 2010) and/or incorporating different observations than discharge in the calibration process (e.g., Thyer et al., 2004; Stisen et al., 2011; Graeff et al., 2012; Corbari and Mancini, 2014; Larsen et al., 2016b; Soltani et al., 2019).
* * *
COMMENT # 2.13

P12L17: Computational resources → should these not be mentioned in the manuscript?

**Answer:**
We added a sentence to the modeling chain subsection to provide the numbers for computational demand

The computational demand was about 0.021 million core hrs for the WRF_SA simulations, 0.32 million core hrs for the WRF-H_SA calibration runs and 0.042 million core hrs for WRF-H_FC on a 2.3 GHz Intel Haswell system.

and a further sentence in the calibration methods section:

The calibration period length of 3.5 months (2015-04-15–2015-07-31, including 14 days of spin-up) is selected as a compromise between the number of model runs (about 2000, which relates to about 0.32 MIO core hours for WRF-H_SA), required during hypercube sampling and PEST optimization, and the available computational resources.

COMMENT # 2.14

P12L20-P13L1: Why/how did you add the baseflow. I see it was necessary, but please elaborate why/how etc. Also – please link the the "_sh"runs in fig 4 + 5.

Related with Comment 1.11.
**Answer:**
In the Ammer catchment, especially in the mountain valleys, glacial overdeepening created considerable aquifers with depths of up to several hundreds of meters, Seiler (1979); Frank (1979). These storage bodies with long residence time withhold precipitation input and act as low pass filters, sustaining the long-term baseflow dynamics. With the conceptual bucket in WRF-Hydro, it is currently not possible to cover these long-term dynamics. It would require either overlaying multiple buckets or a two- or three-dimensional description of groundwater flow, like, e.g., in TerrSysMP (Shrestha et al., 2014). Since we cannot reproduce these long-term baseflow dynamics in WRF-Hydro, we consider baseflow shifting of the simulated hydrographs to evaluate them as if the baseflow deficiencies would not exist. However, the shifting is only applied after the calibration of WRF-H_SA, thus it is purely analytical. The shift amounts are manually derived so that the recession parts of the hydrographs are well resembled. For the subcatchments of the Rott and Ach (Rt-RST, Ach-OBN, Ach-OBH) the baseflow generation is governed by more shallow, quickly reacting aquifer systems and therefore adding constant baseflow would be inappropriate.

We changed the text as follows to explain the motivation and method of baseflow addition in detail. We also linked Figures 4 & 5 at the beginning of the paragraph.

The hydrographs for the calibration and validation periods of the WRF-H_SA runs are presented in Figures 4 and 5. The final parameter sets and goodness of fit measures are listed in Table 3. For all subcatchments, reasonable configurations could be determined. However, for the three Ammer subcatchments (OAG, PEI, WM) it was required to add constant baseflow rates of 2, 4.71, and $5.13 \, m^3 \, s^{-1}$ to the simulated hydrographs, respectively. Adding constant baseflow is justified by the fact that due to the glacial processes of the last ice-age, large storage bodies, that dip reversely with the surface elevation, were formed in the mountain valleys by overdeepening (Seiler,

1979; Frank, 1979). Further downstream, towards the opening of the valley, where the aquicludes reach towards the surface, springs are abundant. Channel inflow from such long-term storage cannot be realized by WRF-Hydro's conceptual bucket scheme. It would require the implementation of a more sophisticated groundwater model. The amounts for constant baseflow are derived manually after the PEST parameter optimization, so that the recession curves of the simulations agree well with the observations. For the Ach and Rott subcatchments channel baseflow is related with shallow aquifers with shorter residence times which should be solely captured by the model.
* * *
COMMENT # 2.15

Table 3: For Am-PEI "coeff"and "expon"the values are larger than the suggest span in table 2. Table 3: For the two parameters mentioned above as well as zmax = 1 (3 instances) and retdp ) = 5, the autocalibration seems to have hit a boundary limit, which could imply model deficiencies. Please mention/elaborate or re-run, using more sound limits.

**Answer:**
First, we would like to point out that in Table 2 the typical parameter ranges are either defined in the NOAH-MP user manual (the LSM parameters) or derived from the authors' experience. In the WRF-Hydro documentation, no ranges are defined for the calibration parameters. The ranges given in Table 2 are not aimed to represent tight limits, which have been exceeded also in other cases than those reported by the referee (e.g., refkdt $< 0.1$). In order to avoid any misunderstanding about the meaning of these 'typical ranges', we removed this information from Table 2 in the revised manuscript.

Concerning the cases where the referee suggests that the autocalibration could have reached a boundary limit, it should be preliminarily recalled that the LH-OAT method was used to obtain a starting configuration for the automated parameter optimization. Then, the PEST method considers some upper and lower boundary limits for the parameters subject to calibration. Several times we experienced that the parameters hit the boundaries that we set, however in all cases we relaxed the constraints, allowing the parameters to exceed the previous limits. In the cases we found negligible (or even null) improvements, we preferred to come back to the previous borders (this typically occurred with lower zmax values, close to 1). To address this, we have put the following statement into the calibration subsection of the manuscript:

During the PEST automated calibration, it occurred that some parameters hit a boundary limit of the calibration range, previously set starting from the results obtained with the LH-OAT method. E.g., the optimized ZMAX values hit the lower boundary for the three upstream catchments. In such cases the constraints were relaxed, allowing the parameters to exceed the previous limits, but if negligible or even null improvements were found, we preferred to come back to the previous borders.

COMMENT # 2.16

P13L10: First sentence: Not understood. Limited – how?

**Answer:**
We have reworded the sentence to make the point more clear. Now it reads:

The commonly favored lumped calibration of WRF-Hydro seems to be rather limited with respect to the transferability of parameter sets among subcatchments and also concerning the numerical efficiency for automated calibration. Especially for complex terrain, e.g., as presented by this study, the distribution of discharge gauges does not agree with landscape units.

COMMENT # 2.17

Table 4-9: I think you should add MAE.

**Answer:**
We agree with the referee and replaced RMSE by MAE throughout the manuscript.

COMMENT # 2.18

P20L6-13: Isn't that just the latent heat results again? They look the same. If not, how are they different and why?
P20L6-9: Unclear which scenario is most realistic? Why not just use the best? Better word than scenario?

**Answer:**
The simulated latent heat and evapotranspiration time series are indeed identical. However the observations for latent heat are obtained from a flux tower and those for evapotranspiration are from lysimeters. We combined Figures 8 with Figure 11, by showing the simulation results, the flux tower and the extensive lysimeter management data together. In WRF, latent heat flux (W m$^{-2}$) is converted to evapotranspiration (kg m$^{-2}$ s$^{-1}$) with the latent heat of vaporization constant (LVH2O=2.501E+6 J kg$^{-1}$). Furthermore, we skiped the intensive lysimeter management scenario (shown in Figure 11 of original submission). The implementation in Noah MP is congruent with extensive management where grass is only harevested once or twice per growing season. Subsection 3.1.4 contains now only soil moisture.

COMMENT # 2.19

P22L3-4: Please see my previous comment on table 3 about the resulting parameters. "...often tuned to unrealistic values..."

**Answer:**
The statement here is not related to our model calibration and parameter ranges. It is a more general statment, based on Koster et al. (2009) who found that the soil representations in land surface models is typically far away from reality.
* * *
COMMENT # 2.20

P22L14-15: Why? Discuss/relate! This is a discussions section.

**Answer:**
The reason for the statement was actually given in the sentence before. Because of the accuracy of the HATPRO measurement, the conclusion for temperature is clear whereas for humidity we cannot be likewise sure about the underestimation by the model. The term "extremely likely" is then probably the wrong formulation. We therefore changed the statement to:

For temperature, the differences between HATPRO and the models are generally much larger than the HATPRO accuracy. For humidity, the mean deviation lies within the accuracy of the measurement. It can be followed that both models overestimate temperature and probably have a tendency to underestimate absolute humidity. As compared to WRF_SA, WRF-H_FC shows reduced deviations for both variables.
* * *
COMMENT # 2.21

P26L19: Please write "(NSE)"after Nash Sutcliffe efficiency.

Related with Comment 3.6
**Answer:**
Due to artifacts in the RADOLAN precipitation data, we have re-computed the 2016 WRF-H_SA simulation and therefore updated the respective figure and passages. We have also added NSE as you suggested. The details about the changes are outlined in the answer for Comment 3.6.
* * *
COMMENT # 2.22

P28L8-9: Why was this the case?

Related with Comment 1.11 and Comment 2.14.
**Answer:**
The detailed explanation for the baseflow shifting and underestimated baseflow is given in the calibration subsection (2.4.6). For the summary we have updated the sentence about baseflow shifting:

> For the Ammer subcatchments (OAG, PEI, WM), due to long-term hydrogeological storage processes that cannot be reproduced by WRF-Hydro, it was required to correct the negative biases of the baseflow.
* * *
COMMENT # 2.23

P29L2: "For the validation period"→ has this been addressed previously?

**Answer:**
Yes, the results for the calibration were discussed in subsection (2.4.6). As we have re-run WRF-H_SA for the validation period due to artifacts in the RADOLAN precipitation data, the context for this passage has changed. For the respective answer, please refer to Comment 3.6 (and Comment 2.21).
* * *
COMMENT # 2.24

P29L9-10: Why is this a problem (the 1h time steps)? If it corresponds to "SA" mode.

**Answer:**
In its fully coupled configuration (WRF-H_FC), the Noah-LSM would call the hydrological subroutines (routing) at every WRF model time-step which would be about every four seconds. With these short time-steps and the high spatial resolution numerical effects lead to incorrect results. Therefore, to get commensurable simulations with all three model variants, it is required to upscale the hydrological simulation part to one hour resolution, similar as used for the WRF-H_SA runs. We tried to explain this better by rewording the sentence to:

> For the fully-coupled WRF-Hydro run (WRF-H_FC), to obtain commensurable quantities for the evaluation, instead of calling the hydrological subgrid functions at every WRF model timestep (4 sec), an hourly timestep identical to that of the WRF-H_SA model was required. The strategy was selected to avoid numerical truncation effects in the overland routing routines that happen when timesteps are in the order of a few seconds and spatial resolution is about 100 m or below (see also section 2.4.4).

COMMENT # 2.25

P29L24: Reference? On alpine foothills soil texture.

Related with Comment 1.15 and Comment 3.4.
**Answer:**
There are only a few references about soils of the northern limestone Alpine region and most of them are in German. However, we found two application studies for our study region, Ludwig et al. (2003); Hofmann et al. (2009). Moreover, a web service of the Bavarian Environmental Agency (LfU) is available at https://www.umweltatlas. bayern.de/mapapps/resources/apps/lfu_boden_ftz/index.html?lang=de&layers=service_ boden_5%C2%A2er%3D4418466%2C5268428%2C31468&lod=6&stateId=b3ed6873-359d-46d7-ad68-73359d56d749. We added the references as follows:

> This does not even rudimentarily reflect the complexity of the northern limestone Alps and foothills (compare Hofmann et al., 2009, for the Halbammer subcatchment of the Ammer), in particular with the high resolutions of 1 km and 100 m of the atmospheric and hydrological model sections. The incorporation of high-resolution continental or global soil maps like, e.g., recently made available by Hengl et al. (2017) and Tóth et al. (2017) could lead to further improvement here.

COMMENT # 2.26

P30L15: I do not like the use of "hopefully". If you had framed the relevance of the study in the introduction, then this paragraph would be more easily written.

Related with Comment 1.16.
**Answer:**
We changed the paragraph to:

The combined approach offers potential to improve also future Earth System Modeling, like also pointed out by Clark et al. (2015). To experience the full momentousness of coupled atmospheric– modeling future studies should be extended to larger regions to cover the scales of atmospheric recycling processes. Also the descriptions of the hydrological processes in the models should be further refined as computational capabilities increase and with more and more detailed data products becoming available.
* * *
COMMENT # 2.27

Minor: Abstract: "Nominal"–use better wording? P12L1: where "the"model. Sounds better to my taste. P26L25-26: "difficulty in reproducing"instead.

**Answer:**
All changed. The sentence in the abstract reads now:

The comparison of classic WRF and fully coupled WRF-Hydro, both using the calibrated parameters from the offline model, shows only tiny alterations for radiation and precipitation but considerable changes for moisture- and heat fluxes.
* * *
With the development of fully-coupled atmospheric-hydrological model especially the improvement in the hydrological part, whether the improvement in land surface process (e.g., hydrological process) can lead to a better simulation of regional water and energy cycle is an issue that is concerned by the whole literature. Fersch et al., presents a very comprehensive research on this issue by comparing the coupled model (WRF-Hydro) with the original one (WRF_SA). Generally, their experiment is well designed, the results are convincing and the research does have added value to deepen our understanding on the performance of fully-coupled atmospheric–hydrological modeling. I think it can be published in HESS after addressing the following comments.

COMMENT # 3.1

Introduction of WRF-Hydro. From the introduction of the WRF-Hydro, it seems that WRF-Hydro improves the physical description of surface flow and baseflow transportation. But do the WRF-Hydro and WRF_SA have the same runoff scheme? Will and how the surface/subsurface lateral flow feed back to the land (e.g., through re-infiltration)? I suggest the author give some introduction on the difference of basic WRF and WRF-Hydro as not all readers are familiar with this model.

Related with Comment 2.6 and Comment 3.9.
**Answer:**
We added a paragraph to the section *The WRF-Hydro modeling system* to make the differences between WRF, WRF-Hydro, and the different model setups clear.

WRF-Hydro (Gochis et al., 2016) augments WRF with respect to lateral hydrological processes at and below the land surface. It adds a surface storage layer where infiltration excess water is stored and subsequently routed according to the topographic gradient once the retention depth becomes exceeded. This is different to WRF where the infiltration excess depicts a sink term. Thus, in WRF-Hydro the surface water can infiltrate gradually, potentially leading to increased soil moisture. Gradient based routing is can also be activated for saturated soil layers, and in case of oversaturation the water will exfiltrate to the surface where it enters the surface storage body and routing process. WRF-Hydro is connected with the planetary boundary layer in the same way as WRF, the lateral water transport at and below the surface is the crucial difference. Further hydrological processes that are implemented in WRF-Hydro without feedback to the atmosphere are baseflow generation and channel routing. The model has two operation modes, stand alone, driven by gridded (pesudo-) observations (one-way coupled) or fully coupled with the dynamic atmospheric model WRF (fully-coupled).

COMMENT # 3.2

Meaning of different parameters. The current interpretation of physical meaning of parameters is distributed. For example, in P12,L12 "The LSM surface runoff scaling parameter REFKDT is globally set to 2e-06 as smaller values would have decreased infiltration to very small amounts."P28,L12 "Also the percolation parameter SLOPE was mainly reduced as compared to the standard value, meaning that a considerable portion of former infiltration excess water needed to be transferred to the bucket-storage to assure good performance for the simulated baseflow". They should be directly introduced in the calibration section. Moreover, I suggest a brief interpretation of the physical meaning of changes of parameter.

Related with Comment 3.5 and Comment 1.9.
**Answer:**
We followed the suggestion of the reviewers and reformulated the calibration methods section to give an overview of all relevant parameters and also explain the physical meanings of the different parameters in more detail. We added this additional block:

Focusing on the values of the calibrated parameters, it can be observed that the surface infiltration parameter REFKDT, as compared to the standard settings in WRF and Noah-MP (e.g., nominal range of 0.5-5.0, according to Niu (2011); 0.1-0.4, according to Lahmers et al. (2019)), is rather low (therefore allowing lower infiltration) for all sub-catchments except Ach-OBH. The associated LSM surface runoff scaling parameter REFDK is globally set to $2e^{-06}$ as smaller values would have decreased infiltration to even smaller amounts. Also, the percolation parameter SLOPE was mainly reduced as compared to the standard values (0.1-1.0, according to Niu et al., 2011b), meaning that a relatively limited portion of former infiltration excess water needed to be transferred to the bucket-storage to assure good performance for the simulated baseflow. As for the surface overland roughness scaling factors, they generally increase remarkably the standard value of 1.0, contributing to the increase of the hydrograph's lag time and the relative reduction of the peak discharge. The retention depth scaling factors, instead, are much closer to the standard value of 1, varying slowly both the total volumes of the hydrographs and the lag time of the initial response of the catchment to rainfall. The bucket scheme parameters should be evaluated considering their mutual influence on the model exponential equation. In general, the higher the ZMAX value, the slower the response time of the bucket, and the higher the COEFF

value, the higher the potential contribution of the bucket model to the total runoff. From this point of view, the most reactive subcatchment is Am-PEI. In this subcatchment, REFKDT is rather small and therefore the contribution by the bucket needed to be quicker also because of the rather large subcatchment site. However, this bucket's behavior cannot be appreciated by looking at the hydrographs in Fig. 4, both because in the simulations the values of the bucket storage height are never close to ZMAX, and because the resulting discharge in the graph also depends on the contribution of the upstream catchment Am-OAG. During the PEST automated calibration, it occurred that some parameters hit a boundary limit of the calibration range, previously set starting from the results obtained with the LH-OAT method. E.g., the optimized ZMAX values hit the lower boundary for the three upstream catchments. In such cases the constraints were relaxed, allowing the parameters to exceed the previous limits, but if negligible or even null improvements were found, we preferred to come back to the previous borders. Finally, it is noteworthy to highlight that, since the study focuses on land-atmosphere exchange, and river routing has no feedback to the LSM, the channel parameters (geometry, roughness coefficient) are not further optimized concerning peak timing.
* * *
COMMENT # 3.3

What is the uncertainty of observations especially for the evapotranspiration? Does the improvement exceed the uncertainty?

**Answer:**
There are two types of uncertainty that come into play here. The first one is connected with the precision and accuracy of the instruments. The second one relates to the scale (spatial support) of measurement and simulations. However, given the accuracies of the sensors and the long term averaging for the diurnal cycles and the performance measures we can rate the improvements to outrange uncertainty. And we see the improvements also for the correlation measures that are not related with systematic over- or underestimation. With respect to evapotranspiration and latent heat flux, both observations state an uncertainty of about 10% (Mauder et al., 2006, 2018). Thus, the improvements seen in the diurnal cycles are largely exceeding the observation uncertainties.
* * *
COMMENT # 3.4

P10, L25: What do the "two dimensional"and "three dimensional"mean? Does the "soil layer depths"mean the "soil layer thickness"?

Related with Comment 1.15 and Comment 2.25.
**Answer:**
Yes, the soil layer depth refers to the soil layer thicknesses. We have reformulated the paragraph, and use the term thickness instead of depth. Also, we removed the terms "two dimensional"and "three dimensional":

Large parts of the modeling domain and the considered river catchments exhibit mountainous terrain with steep slopes covered by shallow soil layers. Here, the model's general assumption of two meter soil thickness (or depth to bedrock) does not hold true as it may lead to overestimated retention of infiltrating water. Therefore, in this study, the soil layer definition was changed from a domain uniform to a grid point based representation and soil layer thicknesses were set to the Noah-MP standard (0.1, 0.3, 0.6, 1 m) distribution for hillslopes below 50 % and to more shallow values (0.05, 0.05, 0.1, 0.1 m) for all slopes >50 %. The 50 % threshold led to a realistic discriminability of valley bottoms and hillslopes.
* * *
COMMENT # 3.5

P12, L12: The LSM sets the REFKDT as 0.2e-6, this value is not within the range given in the Table 2.3. section 2.4.6: Please give the timescale of calibration and validation.

Related with Comment 3.2.
**Answer:**
The REFKDT parameter is erroneously mentioned here. Instead the parameter addressed in this sentence is REFDK which is the scaling parameter for the infiltration equation. We have set this to the documented standard value which is $2e^{-06}$.
* * *
P13,L4-8: If you do not include the abnormally high value, which NSE efficiency you can get? Will it comparable to that in calibration period.

Related with Comment 2.21.

**Answer:**

We investigated the issue with the radar artifacts in the RADOLAN-RW dataset in more detail and identified two periods with erroneous signals. Especially the first one, end of June 2016 had a strong impact, in particular on the upper Ammer catchment. We cross-checked with gauge observations in the region, and also the suspicious cake-piece structure of the precipitation fields led to the decision to set the amounts for these less than 6 hour periods to zero rainfall. Figure 3 shows now the

[Figure]

Figure 3: Re-simulated hydrographs for the validation period, 2016-05-01–2016-10-31 with artifact corrected RADOLAN precipitation data.

re-run of WRF-H_SA for 2016, with updated performance measures. The NSE and VE values improved for all locations, in particular for the Ammer (shifted NSEs for Am-OAG: 0.41 → 0.71, Am-PEI: 0.24 → 0.54, Am-WM: 0.30 → 0.53). For 2015, we did not find any issues with the RADOLAN precipitation data. We replaced the figure in the manuscript with the new version. Accordingly, the following passages were updated:

The performance statistics for the validation period are comparable to those of the calibration period and in the cases of Ach-OBN, Am-OAG, and Rt-RST even improved. For Ach-OBH, as expected due to the disabling of the reservoir option, the buffering effect of the lake cannot be reproduced by the model, thus leading to an overestimation of most of the peak values. However, with respect to the overall discharge in the Ammer catchment, these cut-off peaks are rather small. The performance for Am-WM is lower for both 2015 and 2016, as it aggregates the mismatches of all the upstream subcatchments.

and

For the validation period (2016-05-01–2016-20-31), the skill measures could largely outperform those of the calibration period. However, some structural deficiencies like the underestimated baseflow in the mountainous parts or negative bias remain. It is concluded that some of the processes in the catchment cannot be depicted because of the model physics or the lumped parameter estimation approach.
* * *
COMMENT # 3.7

P26,L8: What do the 50% and 100% mean? Which reference they refer to?

**Answer:**
The percent values are with respect to the WRF-H_FC simulations. The range between 50 and 100% is for the different subcatchment. The sentence was updated to make this clear.

Surface runoff (infiltration excess) with WRF_SA is 50% (for some of the subcatchments more than 100%) higher than with WRF-H_FC and WRF-H_SA.
* * *
COMMENT # 3.8

P26,L17: Just a suggestion. Will the results improve if you use daily streamflow, as the hourly streamflow is difficult to simulate?

**Answer:**
We computed also the daily discharge goodness measures and found that the results do not improve for WRF-H_FC (the fully coupled simulation).
* * *
COMMENT # 3.9

P29,L12: Again, I do not know how does the lateral water transport saturate the soil as you do not introduce this in the model introduction section.

Related with Comment 1.14 and Comment 3.1.

**Answer:**

There are two mechanisms in the overland routing of WRF-Hydro that affect infiltration, surface storage and surface routing. Surface storage keeps the infiltration excess in the model, whereas in WRF_SA this quantity is removed from the model and not available for subsequent time steps. Thus, in WRF-H_SA and WRF-H_FC infiltration excess can infiltrate over multiple time steps. The surface gradient routing further increases infiltration capacity by transporting infiltration excess to adjacent grid cells that may remain without infiltration otherwise. We updated the model description section as follows to make this also clear to the readers. See Comment 3.1 for the updated text.

[revised manuscript text omitted]